# MethPhaser: methylation-based long-read haplotype phasing of human genomes

Yilei Fu [1], Sergey Aganezov[2], Medhat Mahmoud [3,4], John Beaulaurier [2], Sissel Juul [2], Todd J. Treangen [1,5] ✉ & Fritz J. Sedlazeck [1,3,4] ✉

The assignment of variants across haplotypes, phasing, is crucial for predicting the consequences, interaction, and inheritance of mutations and is a key step in improving our understanding of phenotype and disease. However, phasing is limited by read length and stretches of homozygosity along the genome. To overcome this limitation, we designed MethPhaser, a method that utilizes methylation signals from Oxford Nanopore Technologies to extend Single Nucleotide Variation (SNV)-based phasing. We demonstrate that haplotype-specific methylations extensively exist in Human genomes and the advent of long-read technologies enabled direct report of methylation signals. For ONT R9 and R10 cell line data, we increase the phase length N50 by 78%-151% at a phasing accuracy of 83.4-98.7% To assess the impact of tissue purity and random methylation signals due to inactivation, we also applied MethPhaser on blood samples from 4 patients, still showing improvements over SNV-only phasing. MethPhaser further improves phasing across *HLA* and multiple other medically relevant genes, improving our understanding of how mutations interact across multiple phenotypes. The concept of MethPhaser can also be extended to non-human diploid genomes. MethPhaser is available at https://github.com/treangenlab/methphaser.

The emergence of long-read sequencing technologies has enhanced our understanding of the human genome, uncovering novel types of variations between individuals and even tissues[1]. The latest advancements allow us to gain a more comprehensive understanding of single nucleotide variations and more complex structural variations at unprecedented levels. These discoveries reveal novel loci that could potentially impact diseases, evolution, or other important phenotypes[2-6]. Over the past few years, novel computational methods have enabled advancement in all these fields by providing more complete human genomes[7,8], more comprehensive detection of variants at germline and somatic levels, as well as a more realistic view of the genome by providing phasing information[2,9,10]. Here, phasing refers to the assignment of variants to the two copies of each genome as they are present in human and other mammalian cells[11]. Once this assignment is made, it becomes easier to investigate the consequences of two alleles co-occurring on the same DNA molecule, which can have different impacts on specific genes[12]. We differentiate between cis-relationship, which occurs on the same DNA molecule, and trans-relationship, which occurs on opposite DNA molecules. Identifying the relationship between two or more variants is crucial for many downstream applications.

To perform this phasing, which refers to the relationship between variants, three strategies exist. A population-based phasing strategy leverages the information of co-occurrences of variants across multiple hundreds to thousands of individuals[13]. Thus, while this strategy can easily phase entire chromosome arms, it can only do so on common SNPs in the population, and therefore misses the rare and likely disease-causing variants[14,15]. Another strategy, trio-phasing, involves obtaining parental information and using it to phase a single nucleotide variant (SNV) based on the co-occurrence of the parents[16]. This can

[1]Department of Computer Science, Rice University, Houston, TX, USA. [2]Oxford Nanopore Technologies Inc, New York, NY, USA. [3]Human Genome Sequencing Center, Baylor College of Medicine, Houston, TX, USA. [4]Department of Molecular and Human Genetics, Baylor College of Medicine, Houston, Texas, USA. [5]Department of Bioengineering, Rice University, Houston, TX, USA. ✉e-mail: treangen@rice.edu; Fritz.Sedlazeck@bcm.edu

also phase rare SNV in the population and produce chromosome-wide phasing but requires additional sequencing of the parents, which is not always available[17]. More importantly, this strategy fails to phase de novo variants, which are often causative in diseases such as intellectual disability[18] and Mendelian diseases[19]. Lastly, per-read phasing leverages only the linking information of variants that are shared in the same read. While this has the advantage that even a de novo variant can be assigned to a specific haplotype, it is highly reliant on the distance between two heterozygous SNVs and the read length. Thus, if the read length is not sufficiently long to span the distance of two heterozygous SNVs, phasing cannot be inferred. While per-read phasing is the most comprehensive method, it heavily relies on the read length and may only obtain regional phase information, which is referred to as a phase block. Within a phase block region, the phased SNVs are assigned to each haplotype and thus comparable. However, between two phase blocks, the assignment of SNV to haplotype 1 or 2 cannot be determined as no connection information is available. This study aims to improve phasing to be comprehensive, without the need of additional sequencing, while obtaining longer phase blocks and reducing their number.

In recent years, the reduction in cost and increase in yield of long-read sequencing technologies have enabled its use to improve variant detection and to directly improve phasing. Utilizing ever-larger read lengths has resulted in complete phasing as they routinely span the distance of two or more heterozygous SNVs. The average span between two heterozygous SNVs in a human is approximately 1 kbp[5], which long reads easily span. Nonetheless, there are certain regions of the human genome with higher concentrations of homozygous SNV, which can pose a challenge for achieving complete phasing even with long read samples that have an N50 read length of 30 kbp or more.

Another source of information provided by long reads is the methylation signal. Higher or lower methylation is often associated with the inactivation or activation of certain regions of the genome and can be another useful tool for the interpretation of certain genes[20]. Several sequencing methods, like whole genome bisulfite sequencing (WGBS), Hi-C sequencing, and short-read sequencing, have been widely used to analyze methylation signals[21,22]. However, those three methods have several drawbacks too. For instance, bisulfite sequencing is proven to have uneven coverage[23,24], leading to lower-confidence CpG locations. Short reads sequenced from bisulfite-converted libraries often suffer from poor low alignment scores. On the other hand, long-read technologies have been proven to preserve long-range DNA information without the need for massive pre-processing steps[25–27]. Methylation patterns can vary significantly across different tissues, making them tissue-specific. Recent studies have demonstrated a clear relationship between methylation patterns across haplotypes, with some exceptions. One exception is sex chromosomes, where males have only one copy of the X and Y chromosomes, while females have two copies of the X chromosome (i.e., diploid). The activation and inactivation of one of the X chromosome copies is determined by high methylation of one copy[28,29]. This is done at random and doesn't follow the haplotype structure. Nevertheless, in theory, this would mean one can leverage methylation signals across stretches of homozygosity for autosomal regions of the genome. By doing so, there is potential to improve upon the variant phasing and be less reliant on the distance between two heterozygous SNVs.

Oxford Nanopore Technologies (ONT) allows individual DNA molecules to pass through a mutated biological nanopore. The pores are embedded in a membrane, across which a voltage is applied that generates an ionic current through the pores. The detection of modified bases relies on their unique current signatures, which differ from canonical, unmodified nucleotides, as they pass through the pore[21,27,30,31]. In recent years, many tools have been developed to call 5mC at CpG sites, like Nanopolish[27], Megalodon[32], Guppy[33], DeepMod[34], and Remora[35]. ONT has combined Remora into its official state-of-the-

art basecallers Bonito[36] and Dorado[37], and in this study, we are using Remora as our methylation caller. Since the emergence of methylation calling technologies, several methods have been developed for utilizing such information to perform phasing on human genomes, all of which depend on allele-specific methylations. Also, in the same haplotype, base modification probabilities in the same CpG locations are similar. MethHaplo[38] combines allele-specific DNA methylation and SNVs with bisulfite and Hi-C sequencing data for haplotype region identification. A WGBS-based methylation haplotype block identification method[39] was also proposed for improving heterogeneous tissue samples and tumor tissue-of-origin mapping from plasma DNA. NanoMethPhase[40] and PRINCESS[2] are the other two methods that propose to use SNVs and methylation signals together for methylation phasing. Similarly, ccsmeth[41] also provides methylation phasing on PacBio circular consensus sequencing data[42]. However, none of the above-mentioned methods provide a path to enlarge phase block length and further phase more SNVs on ONT data. They all either perform phasing on distinct data types (short reads and Hi-C reads), or only phase methylation events using phased SNV signals. They do not provide improved SNV phasing or read haplotype tagging results.

This paper investigates the utility of methylation signals for the phasing of SNV and general variations. We show that utilizing methylation can improve the phasing overall and thus be leveraged to assign more reads (i.e., shorter reads) to haplotypes, thereby boosting the ability to call variants. We developed MethPhaser, a tool that operates on a set of already phased variants based on SNVs from, e.g., WhatsHap[11] or Hapcut2[4]. MethPhaser then utilizes the heterozygous methylation information across the autosomes to connect phase blocks and thus improve the overall phasing. We showcase the method on HG002/NA24385, where benchmark data is available based on phased long-read assemblies and variant benchmarks. Here we highlight the performance of MethPhaser genome-wide on autosomal chromosomes, with a special focus on medically relevant genes, especially the *HLA* region[43], where phasing is most important. To demonstrate the versatility of MethPhaser, we evaluated its performance on various human populations and tissue types. Our results show that MethPhaser improves variant-based phasing with minimal impact on phasing errors. This represents a novel and valuable enhancement for variation analysis.

## Results

### Haplotype-specific methylation provides insight for phasing

To illustrate the driving force behind MethPhaser, we offer two examples that demonstrate haplotype-specific methylation patterns on autosomal chromosomes. Figure 1 shows how methylation can be used to phase variants across the autosomes. Figure 1a contains the differential median per haplotype methylation levels across one SNV phased block (Chr1: 41,306,391-43,566,623). Here multiple CpG locations (along the *X*-axis) show differential methylation patterns between the two haplotypes. The *Y*-axis is the median of all the scores of this CpG location in each haplotype (blue: haplotype 1, orange: haplotype 2) reported by ONT basecaller (0-1). We can see a clear distinction between the two methylation values (blue and orange) along the haplotypes. Figure 1b is a more detailed example that shows haplotype-specific methylation patterns exist in non SNV based phased regions. In this IGV plot, we can see that 5 CpG locations in the reads cannot be phased by SNV-based methods (marked in red), even though they are known to be haplotype-specific methylations. These two examples clearly show the motivation of MethPhaser which uses methylation as an extension of SNV phasing.

### Overview of the MethPhaser approach

We developed MethPhaser to leverage methylation information to extend SNV-based phasing. This approach is conceptually novel as we leverage heterozygous methylation signals across stretches of

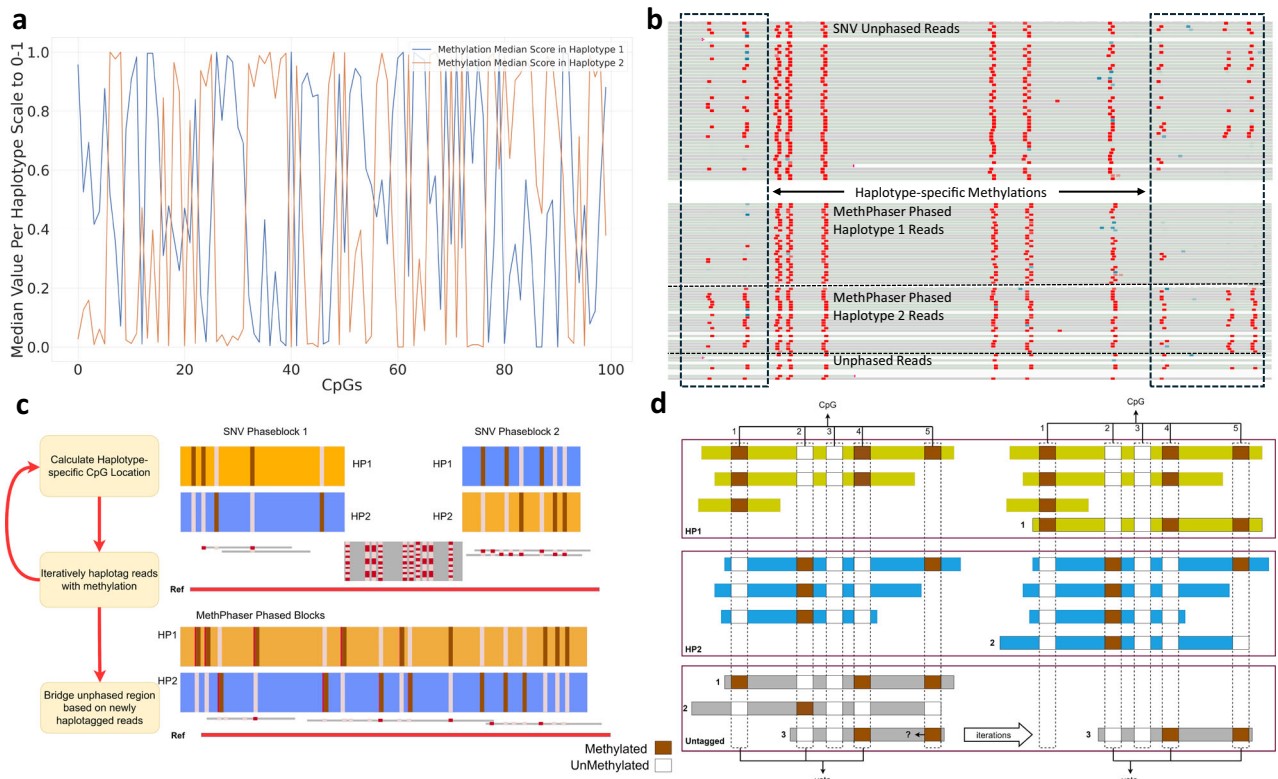

**Fig. 1 | Haplotype-specific methylation in heterozygous regions and overview of MethPhaser. a** Differential methylation signal patterns among two haplotypes in the heterozygous region. Two lines represent the median of reads' methylation scores in each CpGs in a SNV phase block (Chr1: 41,306,391-43,566,623). **b** Example of haplotype-specific methylations on unphased regions from SVN-based phasing at chr6:30,893,180-30,893,730. **c** MethPhaser overview. Left panel: MethPhaser includes 3 major steps: 1. Calculate statistically different methylation CpG locations based on SNV-haplotagged reads. 2. Iteratively haplotag reads with methylation information. 3. Bridge disconnected phase blocks with newly haplotagged reads. Right panel: MethPhaser assign previously unphased reads to haplotypes and later

form a larger phase block. **d** Schematic example of a region where methylation information can help to improve phasing overview. MethPhaser summarizes the methylation patterns in SNV haplotagged reads in each phase block and assigns unhaplotagged reads via pattern matching. MethPhaser locates similar patterns on SNV-untagged reads and haplotypes in phase blocks and assigns matched reads into haplotypes. With newly assigned reads, MethPhaser iteratively updates the existing methylation patterns in the phase blocks and tags more untagged reads. With more tagged reads, the boundary of the phase block can be extended to the unphased region and further close the gap.

homozygous SNV regions for autosomal chromosomes. MethPhaser takes as input a file containing pre-phased SNVs (e.g., from WhatsHap[11]) along with a BAM file containing methylation tags. MethPhaser is described in detail in the method section. In brief, there are three main steps of MethPhaser (Fig. 1c): 1. Calculate statistically different methylation CpG locations based on SNV-haplotagged (i.e., labeled) reads. 2. Iteratively haplotag reads with methylation information. 3. Bridge disconnected phase blocks with newly haplotagged reads. Steps 1 and 2 are repeated until either no more reads in the disconnected regions can be further assigned or the iteration number reaches a user-defined limit (default 10). An illustration of the outcome of these 3 steps is shown in Fig. 1c right side panel. In this example, two regions of the genomes are independently phased based on SNV information but couldn't be connected, either because of a stretch of homozygous SNV or other reasons. Here MethPhaser can combine both phase blocks and thus generate a single larger block by leveraging the heterozygous methylation signal in this region. In addition, MethPhaser tags the unassigned reads to either haplotype to enable a more comprehensive haplotype-specific variant calling. This is particularly important for, e.g., somatic variations, where it is important to understand whether they occur within a haplotype. In the end, MethPhaser produces several outputs: a BAM file with altered haplotype assignment of reads, a vcf file with altered SNV phasing result, and also two files that reveal the phase block relationships: CSV files indicate the relationships between SNV phase blocks, and CSV files indicate previously unhaplotagged reads' haplotype assignment. The overall

pipeline of MethPhaser includes initial SNV-based phasing and validation is illustrated in Supplementary Fig. 2.

Figure 1d shows a schematic example of how MethPhaser works. Five CpG locations (1–5) were identified based on the reference genome and three untagged reads (1–3). We first retrieve the base modification score from the basecaller, and the darker red means the score is high, while the lighter red means the score is low. We use the Wilcoxon rank sum test[44] to determine which CpG locations have a statistically different score between the two haplotypes. CpG location 3 does not have a statistically different score across the haplotype, so it cannot be counted as a vote. The reading assignment is divided into two steps: First, read assignment based on SNV phased reads. For the CpG locations 1, 2, and 4, we access the base modification score on untagged reads and see if the score is closer to either haplotype 1 or haplotype 2. In this example, the untagged read 1's base modification score on location 1 is likely classified into haplotype 1, while untagged read 2 is into haplotype 2. Based on the available votes, untagged reads 1 and 2 can be assigned to haplotypes 1 and 2, respectively. MethPhaser's default parameter requires a minimum of three votes to determine a read's haplotype. At least two base modification scores are required for the Wilcoxon rank sum test[44] (e.g., location 5 at step 1). Therefore, the untagged read 3 could not be assigned in step 1 because of insufficient votes. Second, after one iteration, the number of base modification scores is sufficient at CpG location 5, which makes the vote number of untagged read 3 sufficient. The untagged read 3 can be further assigned to haplotype 1. With read assignment results,

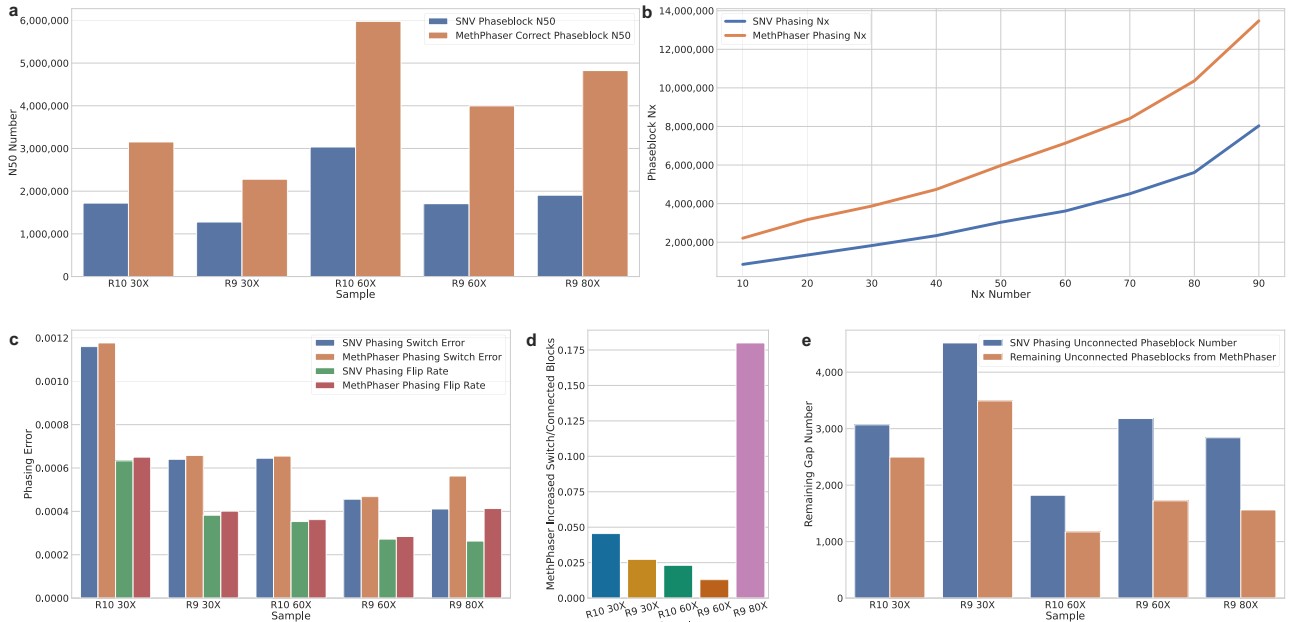

**Fig. 2 | Phasing improvements of MethPhaser across HG002. a** SNV Phase N50 and MethPhaser Corrected N50. MethPhaser increases the N50 of phase block (falsely connected removed) by 1.6–2.5X. **b** N10–N90 curve for MethPhaser on HG002 60X reads. **c** Comparison of SNV and MethPhaser's switch error and flip rate. MethPhaser maintains the same level of switch error as the SNV-based method. MethPhaser increases the flip rate by 0.02% over the SNV-based method. **d** Increased switched SNVs/MethPhaser connected phase block number. Meth-Phaser generates less than 5% phasing error with 30X–60X reads. A higher error rate in 80X reads is observed due to a noisier SNV phasing result. **e** Remaining disconnected gaps on HG002 show that MethPhaser leaves fewer unconnected blocks than SNV phasing methods. With the same coverage, reads from R10 flow cells generally produce fewer gaps.

MethPhaser later perform phase block onnection (Supplementary Fig. 3). MethPhaser takes the reads that are assigned by both neighboring extended blocks into votes for the phase block relationship assignment. With MethPhaser defined extended boundary, the previously untagged reads can be tagged based on the SNV and methylation information in the first and second extended boundaries. The read is assigned to a switched haplotype in those two boundaries, which indicates the switching relationship between those two neighbor SNV phase blocks.

## MethPhaser: Methylation as an extension of SNV phasing

Next, to assess the accuracy of MethPhaser, we utilized the Genome in a Bottle (GIAB) SNV benchmark (V4.2.1) and its reported phasing information based on assembly[45]. Here we compared the benchmark phasing results to those obtained using standard SNV-only phasing and the MethPhaser-enhanced phasing using ONT 60X reads from R9 flow cells. We assessed the performance of the SNV and MethPhaser output based on N50 (i.e., length of phase blocks) and phasing errors. We measured these errors as either flip errors (i.e., single SNV assigned to the wrong haplotypes), switch errors (i.e., all subsequent SNVs are consistently assigned to the wrong haplotypes), or hamming distances (i.e., total differences of SNVs) based on WhatsHap compare (see methods for details). MethPhaser was able to extend the N50 phasing information from the standard SNV phasing (see methods) approach by 1.6X times. For the entire HG002 autosomal genome, MethPhaser reduces the number of gaps between the phaseblocks (i.e., the continuous region where the relationship between the SNVs is reported) from 3179 to 1722 which represents a significant improvement from 1,706,719 N50 to 3,997,227 N50 of phase length. This includes an increase in the flip rate of 0.02% and no switch error increase compared to the SNV phasing alone. The main error mode from Meth-Phaser is more likely to be switch error due to assigning blocks of phasing wrongly, resulting in switch errors or flipping. As we show for HG002 R9 60X reads, Methphaser exhibits a subtle switch error increase of 0.0012% when generating significantly longer phase blocks.

To further explore how much error was introduced by block connection, we calculated the newly introduced switches per phase block connection by MethPhaser. With HG002 R9 60X reads, the newly introduced switch errors per MethPhaser phase block connection is around 1.3% (Fig. 2d).

We next investigated if MethPhaser requires 60x coverage or can produce similar advancements on different coverage levels. Figure 2a shows the N50 comparison based on R9 data across 30, 60, and 80x coverage. Here, 30x represents a single flow cell run. In each case, we could report a significant improvement of SNV phasing based on methylation signals for 30x (1.78 fold), 60x (2.34 fold), and 80x (2.53 fold) coverages. To further show MethPhaser's improvement in phase block length, we also tested the phase block N10–N90 (Fig. 2b). In addition, we also measured the increase in switch error rate, which was from 0.0010% to 0.0152% across the different coverage data sets, as shown in Fig. 2c. Altogether this leads to many more phased SNVs, and thus, the number of unphased regions (i.e., gaps) is reduced. Overall MethPhaser can reduce the gap percentage from 81% (R10 30x coverage) to 54% (R9, 60x coverage) compared to the previous SNV-based gaps between phase blocks (see Fig. 2e). These results show a clear improvement independent of the coverage levels.

Next, we investigated the performance of R10 flow cells from ONT. These represent significant improvements for SNV calling and, thus, potential improvements in SNV phasing itself. We observed that the phasing N50 indeed increases with R10 compared to R9 (see Fig. 2a, ~3000 kbp vs ~2000 kbp). Nevertheless, Methphaser is still able to improve upon SNV-based phasing on R10 data by increasing the N50 by around 1.5 times. Even with only 30X coverage, our program achieves a higher phase block N50 (3,152,506) compared to 60X SNV phasing (3,034,229) alone. Figure 2c shows the consequences of a small switch error increase (by 0.0532%) compared to SNV phasing.

To provide a specific example of why this matters we will now discuss the Thiopurine methyltransferase (*TPMT*) gene. *TPMT* encodes the enzyme metabolizing thiopurine drugs[46] and includes SNV sites, located 8 kbp apart, that can lead to reduced functionality of *TPMT*.

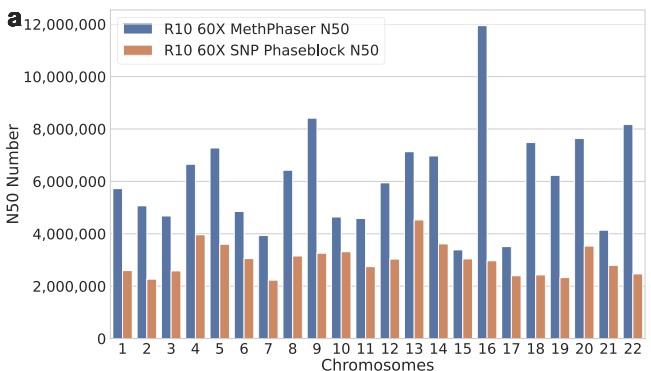
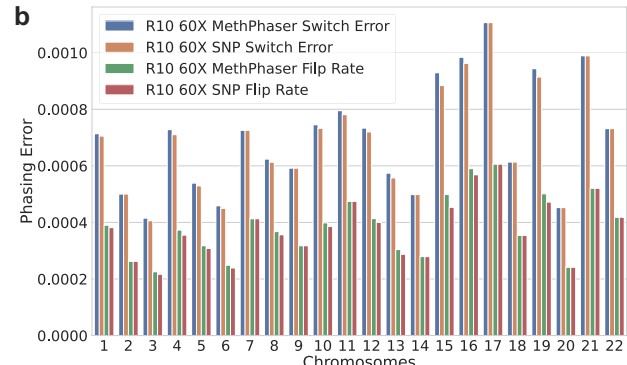

**Fig. 3 | Phasing improvements per chromosome.** Per Chromosome N50, the switch error and flip rate of R10 60X reads. **a** Improvement of phase block N50 on each chromosome. **b** MethPhaser maintains the same switch error and flip rate level.

Given the co-occurrence of SNV on both sites, it is important to determine if they are in cis-relationship, meaning *TPMT* function is reduced, or in trans-relationship, which would result in a deactivation of *TPMT*[47,48]. The latter has severe implications for a patient and thus could lead to his/her death by administering medical treatment. Given this motivation, we investigated the ability of SNV and methylation-based phasing to obtain the correct results for *TPMT*. We identified differences in the ability of R9 vs. R10 flow cells to phase this important gene entirely. MethPhaser was able to connect the phase blocks on R9 and thus close the gap, leading to a fully phased *TPMT* gene based on the methylation information. Supplementary Fig. 1 shows this across phase blocks based on the IGV image together with the gene annotation.

Independent of the flow cell and error rates of Nanopore sequencing (R9 vs. R10), MethPhaser shows significant improvements by closing the unphased regions and connecting SNV phase blocks (Fig. 2e), thereby extending phasing across large regions and connecting genes into one continuous phase block. When we extend this observation across the entire genome, we indeed see this effect. Figure 3 shows the N50 increase, switch error, and flip rate of Meth-Phaser on each chromosome. In Fig. 3a, we see that MethPhaser increases phase block N50 on each chromosome, except chromosome 15. During the experiment, we discovered there is no SNV phase block before the centromere region on chromosome 15, which reduces the N50 increase. Figure 3b shows again that the increase results only in a small increase in phasing errors. The largest increase in switch error is 0.0045%, and the largest increase in flip rate is also 0.0045% compared to SNV-based phasing. Thus highlighting that the N50 improvements do, on average, not increase the error rate significantly.

We measured the CPU runtime of MethPhaser based on the HG0002 R10 60x data set. Overall MethPhaser required 23:20:29 total wall time which equals 222.4 CPU hours with 64 threads specified on an AMD EPYC 7742 64-core Processor.

**MethPhaser improves insights into complex medical genes**
Building on our previous genome-wide results and taking into consideration the challenging but medically relevant nature of *TPMT* (thiopurine S-methyltransferase) gene, we sought to evaluate the performance of MethPhaser on more medically relevant regions. GIAB has recently published a list and benchmark of 273 medically relevant genes that pose significant challenges to resolve[49]. To determine whether MethPhaser's improved phase block N50 results in a greater number of phased genes compared to SNV-based phasing methods, we conducted a comparative analysis of phase block coordinates using MethPhaser, HapCUT2, and the challenging medically relevant genes (CMRGs) benchmark. The comparison result indicates that Meth-Phaser is capable of phasing a greater number of medically relevant genes together, thus allowing for a deeper insight into the counterplay

of their respective variants. This is exemplified as we could reduce the phase blocks across the 273 genes. Overall, MethPhaser could report phasing for 265 (97.1%) of the genes, while SNV phasing was only reported across 258 (94.51%) of these genes. Furthermore, MethPhaser was able to report the phasing with only 140 phase blocks across the entire set of genes compared to 160 phase blocks from SNV phasing alone. The main contribution of MethPhaser in this CMRG example is joining more genes into single-phase blocks.

To further showcase the importance of MethPhaser and give concrete examples where phasing further matters, we investigated the *HLA* region on chromosome 6. This is a highly complex region of the human genome encoding multiple disease-relevant genes impacting the immune system, diabetes, cancer progression, and many other diseases or general medical phenotypes[50,51]. Thus, a complete phasing helps to interpret the variations across these complex genes of classes I and II.

Across the region, MethPhaser was able to extend the phase block length by 132,079 bp while connecting 3 SNV phase blocks. Meth-Phaser successfully reported larger phase blocks (validated with GIAB trio-phasing result) that connect both genes. Figure 4 shows the example of chr6: 30,400,000–31,300,000, which includes the *HLA-E* and *HLA-C* genes in the *HLA* Class I region. We can see in Fig. 4b that the SNV-based method cannot form a phase block that connects those two *HLA* regions, making the phasing status of the *HLA-E* and *HLA-C* unlinked. However, in Fig. 4c, using the methylation information, we can link *HLA-E* and *HLA-C* together. The IGV plot shows that previously untagged reads are being tagged by MethPhaser and assigned to haplotype 1 or 2. This also enables improvement in the assessment of variants, as exemplified by a haplotype-specific insertion. A more detailed view of the haplotype-specific insertion is illustrated in Supplementary Fig. 4, which includes the methylation levels around the insertion. The reads with that insertion are mostly clustered into haplotype 1 in this example. Thus, MethPhaser is able to utilize methylation to put together two distinct SNV phase blocks and thus extend the phasing throughout the locus without the need of additional data.

Thus overall, we could highlight the importance of MethPhaser and its improvements for phasing not only genome-wide but also more focused on medically relevant genes and regions (e.g., *HLA*) of the human genome.

**Improved phasing over human population and patient data.** In the previous sections, we emphasized the critical importance of phasing and the performance of MethPhaser based on HG002, a cell line where we have benchmark data available. To further validate the robustness and generalizability of MethPhaser, we sought to evaluate its performance across multiple samples from diverse human populations. Furthermore, we wanted to assess the ability to improve phasing even

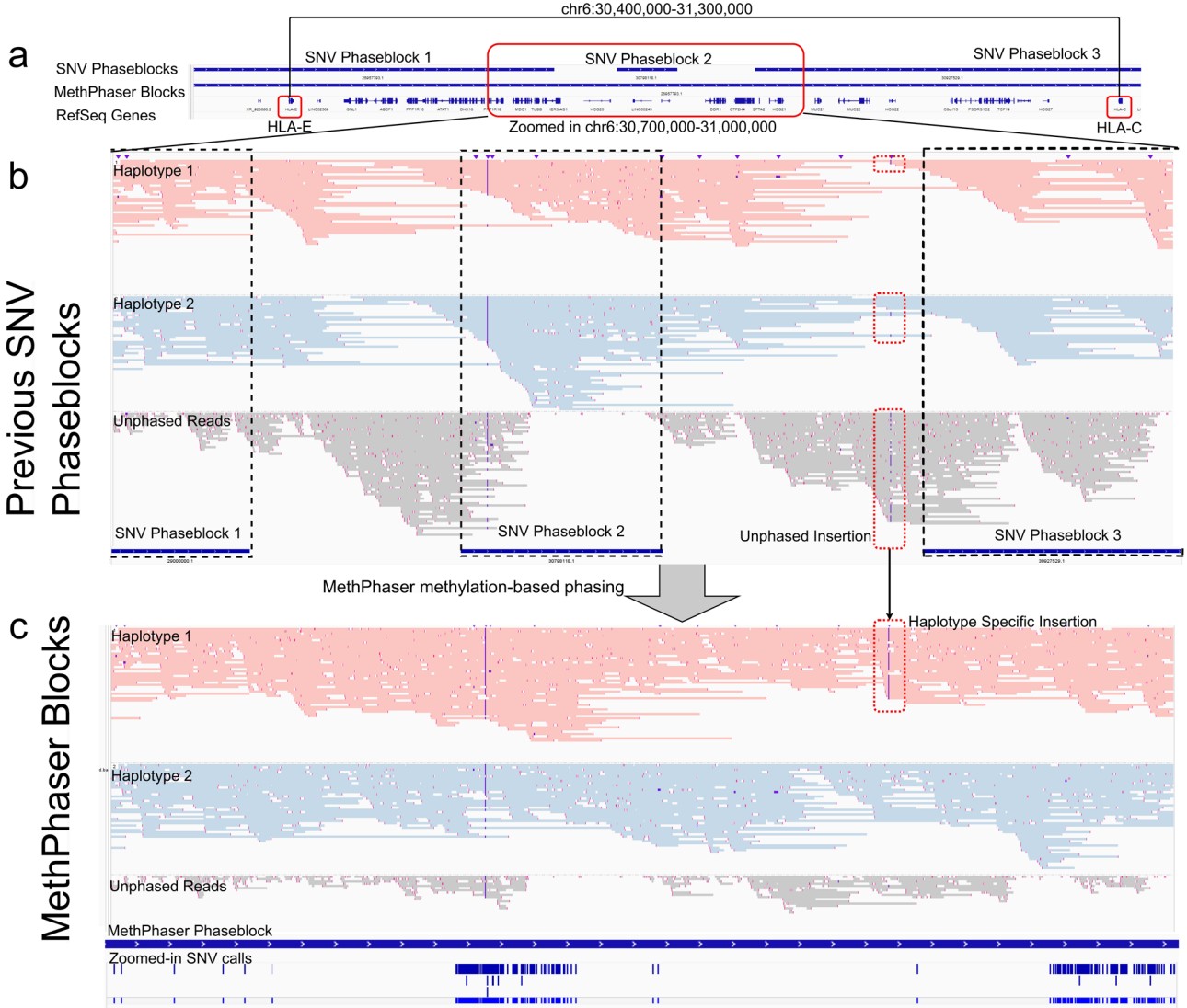

**Fig. 4 | Phasing improvements across *HLA*. a** IGV coordinate of the improved phasing regions. This example shows chr6 30,400,000-31,300,000, which includes two HLA regions. **b** Improved reads assignment connects *HLA-E* and *HLA-C*. The example of MethPhaser improves phasing and read tagging on *the HLA-E* and *HLA-C* genes from the *HLA* Class I region (chr6: 30,700,000–31,000,000). The traditional SNV-based methods can only form three separated phase blocks (with R10 60X ONT reads) that disconnect the two *HLA* regions. The SNV phase blocks are highlighted with boxes with black dotted lines. **c** With MethPhaser, we can achieve a single block that covers two unphased regions. With a closer look at the SNV unconnected regions (chr6: 30,700,000–31,000,000), the IGV plot shows that previously untagged reads are now haplotagged by MethPhaser. MethPhaser also reports a haplotype-specific insertion, which is indicated by the red dotted box. MethPhaser can flip the haplotype assignments of reads in the SNV phase block 2 because it infers that the neighboring phase blocks have switched haplotype assignments.

for less optimal ONT data (e.g., shorter reads). To test the effectiveness of MethPhaser across different populations, we investigated its performance across HG01109 (Male, PUR), HG02080 (Female, KHV), and HG03098 (Male, MSL)[52]. We followed the same analysis procedures that were used for HG002 to enable comparability (see "Methods"). Despite a larger read N50 for these samples (~ 40 kbp) compared to HG002 (~ 30 kbp), we observed similar improvements in MethPhaser across all three samples.

Genome-wide SNV-based phasing reported an N50 phase length between 9 Mbp and 28 Mbp, achieving a higher overall N50 than HG002. This is expected, given the longer read N50 for these samples. Nevertheless, MethPhaser improved upon these N50 phasing lengths in each case, showing an improvement of between 1.69 fold (HG03098) up to 2.55 fold (HG02080) across the samples (Fig. 5a). This is based on the connection of several phase blocks, thereby reducing the number of phase blocks genome-wide. The most extreme example across the three individuals was HG03098, where MethPhaser reduced the number of phase blocks from 947 to 643. Furthermore,

the inclusion of methylation information reduced the number of phase blocks in HG02080 from 1087 to 743. Full details for each sample can be found in the Supplementary Data 3 and 4.

We have further checked the minimum number of phase blocks that covers all phaseable medically relevant genes. In HG01109, we could lower the number of phase blocks to 76 from the initial 82 blocks based on SNV phasing. Two better examples are HG02080 and HG030989, which reduced the phase block number across the medical genes from 123 to 113 and 108 to 96, respectively. Given the much longer read length (HG01109 N50: 46,836, HG02080 N50: 41,591), MethPhaser can cover all GIAB-reported complex medically relevant genes on the genome with HG01109 and HG02080 samples, while the SNV-based method failed to do so on HG03098 with the *RHCE* gene on chr1 (MethPhaser can phase) and the *SRR* gene on chr17 (MethPhaser failed to rescue). To assess the accuracy of MethPhaser's phasing on the HPRC sample, we also performed an evaluation on HG1109 chromosome 1. In this evaluation, MethPhaser only wrongly assigned one block, which leads to almost no increase in switch error and flip

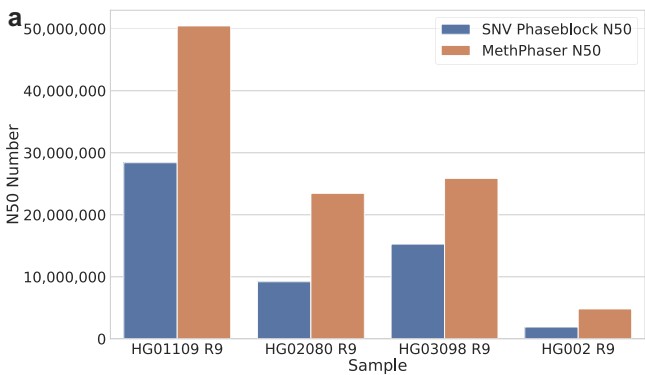
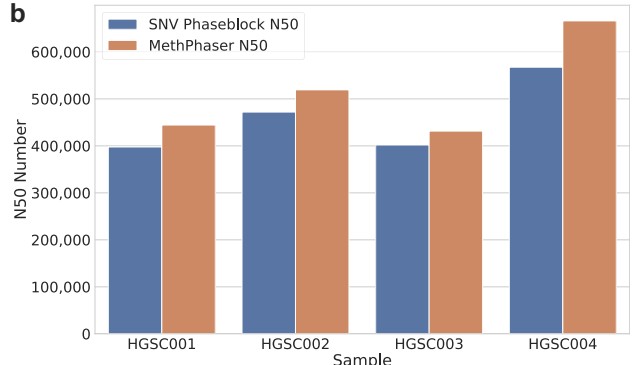

**Fig. 5 | Phasing improvements across different human populations and blood derived samples.** N50 increase of HPRC pangenome samples and patient blood samples. **a** N50 increase in ultra-long, high-coverage pangenome cell-line samples from different ethnic backgrounds. MethPhaser achieved a 1.69–2.55x N50 increase. **b** N50 increase in patient blood samples from different ethnic backgrounds. MethPhaser achieved a 1.07–1.17x N50 increase due to a much shorter read length.

error. However, it increased the hamming distance by 5% as all the SNVs in that block are switched (Supplementary Data 2).

Finally, we wanted to assess the performance of MethPhaser on patient samples (HGSC001-004) that might not be as ideal as certain cell lines. For this demonstration, we sequenced two Hispanic and two European samples with ONT R9 flow cells. Each sample was sequenced with one ONT flow cell, resulting in 24x to 37x coverage. Given the nature of the samples, the N50 is much lower than what was achieved using cell lines (R9 cell line HG002: ~30 kbp, R9 cell line HPRC: ~40 kbp, R9 HGSC: 12 kbp). Thus, the resulting SNV phasing is overall reduced. Across the four samples, we measured an average N50 of 459.7 kbp based on SNV phasing. This was improved from MethPhaser up to 515.2 kbp average N50 (Fig. 5b). Given the reduced initial SNV phase block size, MethPhaser was still able to improve the overall phasing. The degree of improvement was limited by multiple short SNV phase blocks that do not allow us to confidently assign reads between methylation and SNV-based phasing. To further investigate the impact of read lengths on the performance of MethPhaser, we subsampled the read length for R10 30X reads. Supplementary Data 6 shows that by reducing the read length to half N50 (from 27,701 to 13,676), the phasing ability for SNV but also for MethPhaser drops. Thus this highlights that MethPhaser can also improve phasing over blood samples, where the presence of potential tissue heterozygosity could impact the methylation signal.

Despite the modest increase in phase block N50, we were able to significantly reduce the number of total phase blocks from 11,916 to 9837 using MethPhaser. This reduction in the number of phase blocks has important implications for downstream analysis, as it allows for a more streamlined and accurate identification of disease-relevant variants.

## Discussion

Phasing is an important step for obtaining a more complete picture of genetic variation in the human genome, with about 1–5% of human genes being influenced by these unbalanced DNA sequence variants[12]. In this manuscript, we present a novel method, MethPhaser, that utilizes allele-specific methylation information to improve SNV phasing. This is done by bridging the gaps introduced by stretches of homozygous SNVs that otherwise cannot be overcome with SNV phasing alone. To showcase this novel approach, we have applied it to multiple cell lines from different human populations and tissue samples including blood-based samples. The method also shows the potential of being applied to other diploid mammals. In each case, MethPhaser was able to not only extend the SNV-based phasing by joining neighboring phase blocks but also haplotagged more reads that could not be tagged by SNV-based methods, which has the potential to improve

variant calling. By benchmarking against HG002, which has a known truth set, we could demonstrate that this joining of phase blocks leads to only a minimal increase in phasing error (i.e., switch error). This is due to checks and thresholds used by MethPhaser to uphold an accurate phasing result, rather than just greedily joining phase blocks together. The incorporation of methylation information comes at no additional costs, as long reads such as Oxford Nanopore Technologies include methylation information without additional preparation or sequencing runs. This is in contrast to for example Hi-C or Strand-Seq[26] data sets that require additional work and sequencing. Nevertheless, if these extra data sets can be produced they often lead to improved phasing results than long reads alone[53]. Previous work also shows improvements in the phasing of common variants utilizing population data[54,55]. However, this can be limited due to the availability of large population variant catalogs that enable LD calculations. In contrast, MethPhaser is independent of these population data and can also be applied outside of human populations and even to diploid organisms that are non-model organisms. This enables the improvement of phasing provided that methylation is available in the organism of interest. Nevertheless, in most cases this signal can be obtained by either ONT or PacBio HiFi reads[56], highlighting MethPhaser's potential for PacBio HiFi data sets.

The utilization of methylation for phasing is not without limitations, the most obvious of which is the inability to improve phasing for human sex chromosomes in clinical samples. This is because the random deactivation of chromosome X in females would lead to an inconsistent haplotype pattern, where the human cell line is an exception for example in clonal female lymphoblast cell line GM12878 the inactivated X chromosome (Xi) is always the paternal allele[57], also the organisms with skewed X inactivation can theoretically benefit from MethPhaser. Thus, we exclude the X and Y chromosomes from our benchmark. For autosomes, we rely on initial phasing results based on SNVs. MethPhaser relies on initial phasing accuracy based on established SNV phasing methods. This typically is warranted by the quality of long reads and existing methods such as HapCUT2 etc. However, if this is not the case and SNV phasing methods produce high phasing errors (e.g., very high hamming distance in R9 80x results) it will negatively impact the accuracy of MethPhaser. The current version of MethPhaser gives the SNV phasing the benefit of the doubt; in future versions, we will explore alternative approaches to be more robust to high levels of SNV phasing errors. Our results, as well as SNV phasing results, are dependent on the accuracy of methylation detection, overall read length and a certain minimum coverage of the data set. This can be easily seen across the cell lines from HPRC, which had a larger read length N50 (~40 kbp) compared to the blood-derived patient data (12–15 kbp N50). In both cases, MethPhaser could extend

and improve the overall phasing, but this was limited in the patient samples by initially smaller phase blocks. The smaller phase blocks do not provide enough information to make a clear decision on how they should be phased (e.g., Figure 1) and thus MethPhaser does not incorporate them due to the desire to retain a high precision in the phasing accuracy. We further investigated the role of sequencing error or noise in the ability of phasing improvements. Here, we measured the performance of MethPhaser across R9 and R10 flow cells. The latter improves the SNV variant accuracy and thus the phasing overall. Using MethPhaser we could demonstrate the improvement in runs using both R9 and R10 flow cells. Overall, we benchmarked MethPhaser across different conditions, showing that its principles are valid and that it can improve phasing across many medically relevant genes, including the *HLA* region. It is also worth noting that MethPhaser helped the phasing of a haplotype-specific insertion around the *HLA* region in our example (Fig. 4). We include a detailed illustration of the methylation states around that insertion to show that MethPhaser is able to utilize the haplotype-specific methylations to improve read phasing (Supplementary Fig. 4). SNV-based phasing methods like HapCUT2 and WhatsHap perform poorly around the SV regions[58], which is likely to be the reason for the phase block gap located before the *HLA-C* region. MethPhaser shows the potential of phasing haplotype-specific SVs, but requires more experimental results to show the overall improvement, which could be a future step of the benchmarking. We have also compared the impact on MethPhaser's performance with HapCUT2 and WhatsHap's SNV phasing. We observed improvements using MethPhaser for both phasing methods (Supplementary Data 8).

An interesting point when using methylation signals for phasing is the tissue-specific nature of methylation signals. This makes it hard to predict the performance of MethPhaser across different tissues, as the signals can vary. We have tested MethPhaser across different cell lines and in blood, the results of which suggest that its performance is more driven by read length and sequencing coverage than other factors. Even in the most diverse tissue (blood) that carries cells from other tissue types, MethPhaser improved the phasing results over the SNV-based method. Sequencing a certain tissue type alone (e.g., muscle or skin) should thus only improve the phasing results. In blood, while other tissue types might be present, the overall concentration per tissue is minimal. MethPhaser relies on a majority vote to make decisions about combining existing phase blocks, which is often rather conservative but leads to fewer errors as we can show in the benchmark (Supplementary Data 2). In future work, we plan to leverage and improve the detection of heterogeneous signals in the methylation data to improve phasing also for cancer genomes where larger amplifications of regions will carry different methylation signals. Here, MethPhaser could further significantly improve our phasing and thus understanding of cancer evolution.

Overall, MethPhaser is a novel approach to utilize 5mC methylation signals to improve phasing and thus delivers more key insights into the co-occurrence of mutations across medically important genes.

## Methods
The study was approved by the Baylor College of Medicine (BCM) Institutional Review Board (protocol number: H-43884). Blood samples were collected in PAXgene Blood DNA tube (PreAnalytiX, Becton-Dickinson) or Hemogard K2 EDTA tubes (Becton-Dickinson). DNA was isolated with PAXgene blood DNA kit (PreAnalytix, QIAGEN) or Gentra Puregene Blood kit (QIAGEN). Extractions were done following manufacturer instructions in the HGSC CLIA certified laboratory. DNA was quantified using Quant-iT PicoGreen dsDNA Assay kit (ThermoFisher) and its quality estimated by electrophoresis. For ONT sequencing, three-four μg DNA was sheared using g-tubes (Covaris Part number 520079). The g-tubes were centrifuged at 4000 rpm. The sheared sample was bead purified using AMPure XP beads (Beckman Coulter) and resuspended in 50 μL nuclease free water. 1 μL of this eluate was run on the Agilent 2100 Bioanalyzer using the DNA 12000 chip to determine the average size of the sample. The expected average size was 15–20 kb. 47 μL of this eluate was used as input for the ONT SQK-LSK110 library preparation kit. End repair/damage repair and adapter ligation were performed as per the manufacturer's instructions. Final libraries were quantified using the Qubit dsDNA quantification broad-range assay (Thermo Fisher Scientific). Written informed consent was obtained for the collection of blood samples from human research participants, and for sequencing of these samples, and the dataset generation complies with all relevant ethical regulations

### MethPhaser: phasing based on methylation signal
The primary input for MethPhaser consists of a VCF file containing phased SNVs and a tagged BAM file containing methylation information. To perform an extended phasing based on SNVs, three main steps of MethPhaser are taken: 1. Identify haplotype-specific methylation in each SNV phase block with SNV haplotagged reads. 2. Iteratively assign unphased reads in extended SNV phase blocks based on methylation signals in the unphased reads. 3. Infer the relationship between neighboring extended SNV phase blocks with methylation haplotagged reads. The CpG probability reaches beyond the phase blocks, and to make sure that we assign enough reads, we expand our target boundary from the end of the last SNV phase block to the start of the next SNV phase block. We define SNV phase block as $B = \{b_1, b_2, ... b_n\}$, and the start and end of the SNV phase block is $S = \{s_1, s_2, ..., s_i, ..., s_n\}$ and $E = \{e_1, e_2, ..., e_n\}$. The extended phase block, $Be = \{be_1, be_2, ..., be_i ..., be_n\}$, start and end of extended phase block will then be $Se = \{s_1, e_1, ..., e_{i-1}, ... e_{n-1}\}$, $Ee = \{s_2, s_3, ..., s_{i+1}, ..., s_n, e_n\}$.

### Infer haplotype-specific methylation in extended SNV phase block with SNV haplotagged reads
Our method begins with determining whether each CpG location in an extended SNV phase block is haplotype-specific. This involves collecting base modification scores from SNV haplotype-tagged reads and calculating base modification probabilities for each CpG location, as shown in Fig. 1d. The probabilities are the number of MM tags reported from Remora/Dorado called BAM files. We use the Wilcoxon rank sum test[44,59] to determine whether the two allele's base modification probabilities are statistically different (*p-value* < 0.05). Subsequently, MethPhaser generates a list of CpG locations with statistically different base modifications between haplotypes in the same allele, which are considered allele-specific methylations. Due to the fact that those allele-specific methylations are located in the same extended SNV phase block, those are naturally considered haplotype-specific methylations.

### Iteratively assign untagged reads in extended regions
We further assign the reads that cannot be phased by the SNV-based method by collecting their base modification probabilities in pre-calculated allele-specific CpG sites in step 1 and to determine if the probabilities are closer to either haplotype. To ensure better accuracy, we set a minimum coverage of each haplotype (default 3), and a minimum number of votes (default 3) are required for a read's haplotype assignment. For instance, in Fig. 1d, in the unphased read 1 that the first CpG site has a similar probability to the haplotype 1, also the second CpG site. Those CpG locations are treated as votes, and if more votes that above the threshold within one read support it as either haplotype, MethPhaser assigns the read as the voted haplotype. In the same example, MethPhaser disregard the third CpG site on unphased read 1, as it has minimal difference between the two haplotypes (i.e., not statistically different from Wilcoxon ranksum test), and the last CpG site, which lacks sufficient coverage of haplotypes 1 and 2. Based on the three remaining votes, they all suggest haplotype 1, so

MethPhaser assigns the unphased read 1 to haplotype 1. Similarly, it assigns the unphased read 2 to haplotype 2.

After the read assigning with SNV-phased reads, MethPhaser assigns unphased reads in unphased regions iteratively. In the previous example (Fig. 1d), after MethPhaser assigned unphased reads 1 and 2 into haplotype 1 and 2, we can see that the unphased read 3's haplotype was previously unable to decide since there was not enough information on its last CpG location. However, the coverage of haplotypes is sufficient due to our newly haplotagged reads (Fig. 1d, first iteration), and thus, un-haplotagged read 3 can be assigned. Moving on, we update the list of haplotype-specific methylation and haplotagged reads to assign unphased reads further until no more reads can be assigned or the iteration number reaches a user-defined number.

### Infer neighbor SNV phase blocks' relationship from meth-phased reads

Finally, with all these newly haplotagged reads, we can infer neighbor SNV phase blocks' relation from our MethPhaser haplotagged reads. No haplotype-specific SNV information in the unphased regions leads to consistent haplotype assignments in neighboring regions, as shown in the example in the supplementary Fig. 3. However, if those previously un-haplotagged reads are haplotagged by both of the two neighboring extended phase blocks with the SNV and methylation information, they can bridge those SNV-based phase blocks together. The task is to look for reads assigned to haplotypes via methylation data across inbound extensions of consecutive regions. If the haplotype assignment of the same read is switching, MethPhaser will record a vote of the switching relationship between two neighboring phase blocks. By default, Meth-Phaser ignores the largest gap between the SNV-based phase blocks, which is the centromere region, to save computation time.

### Post-processing and result filtering

To reduce the false positive rate, we also applied a filtering script that allows users to determine the minimum reads and voting confidence (the difference between the reads supports "same" and "not same") that support the relationship calls. We provide two parameter suggestions based on the experimental results: 1. Best success rate of connecting SNV phase blocks, with no limitation of minimum reads and voting confidence; 2. Best accuracy, with read coverage/(genome-wide block number/1000) as a minimum read number supporting block relationship assignment requirement and more significant than 0.5 voting confidence ((Votes for one haplotype-votes for the other one)/total votes > 0.5). MethPhaser default uses the best accuracy parameter. The details of the benchmarking process are described in the benchmarking section.

### Output files

The output files contain multiple CSV files that indicate the neighbor relationships and previously un-haplotagged read assignments. Based on those two CSV files, we further modified the BAM and VCF files with pysam[60] and samtools[61] for the user.

1. **BAM file**. The generation of the BAM file depends on the two CSV files mentioned above. Given the neighbor relationship CSV files, MethPhaser generates a list of extended blocks that need to be flipped, i.e., switch haplotype 1 or 2 assignment. Three types of reads need to be processed: **a.** The reads that tagged by SNV-based methods. Those reads are flipped if they are in that list; otherwise, they will be output without modification. **b.** The reads that were tagged by MethPhaser. Those reads are output with their new haplotype assignment, and if the block is in the block-flipping list, the haplotype assignment is switched during the output. **c.** the reads tagged by MethPhaser but overlapped with the previous block. Those reads are removed in this block's read assignment process since they've already been output with the previous block. The read assignments are from the read assignment CSV files.

2. **VCF file**. Similar to the BAM file, the VCF files are generated by considering the neighbor relationship CSV files. MethPhaser reuses the block-flipping list to determine which block's SNVs' haplotype assignments need to be switched.

3. **Neighbor relationship CSV files**. Neighbor relationships are stored in CSV files, which are separated by chromosomes. In the CSV file, the header indicates the SNV and extended phase block, their next SNV extended phase block, and their relationship. The program outputs the true relationship if the truth-phased VCF file is given. Also, the program outputs the read number that supports such relationship assignments.

4. **Previous un-haplotagged read assignment CSV files**. This file is separated by each extended phase block. Each CSV file indicates the previous un-haplotagged reads from SNV-based methods' new haplotype assignment.

### Benchmarking of phasing performance

The benchmarking process includes calculating phase block N50, examining the correctness of neighbor SNV phase block connection, and calculating the haplotagged reads' number. Supplementary Fig. 2 shows an overview of the MethPhaser benchmarking pipeline.

The phase block N50 is calculated as the minimum phase block length, where the sum of its phase blocks with all larger phase blocks spans ≥ 50% of the total phase length[55]. To make sure the phase block connection is as accurate as possible, we chose our "high accuracy" parameter setting for block connecting. The regions that are not correctly connected by MethPhaser should not be considered for the N50 increase. Therefore, When we are calculating the N50 increase, we removed the larger phase blocks that we incorrectly connected based on the comparison of the SNV-based phasing method and the trio-phased VCF provided by GIAB. We compared the correctly connecting N50 with the final phase blocks produced by HapCUT2.

The haplotagged reads are calculated with SAMtools, which provides the ability to count the number of reads with certain tags. In our study, haplotagged reads are tagged with the "HP" tag, and we compared the number of reads with that tag before and after MethPhaser. Furthermore, to show MethPhaser can haplotag reads in homozygous regions, we also visualized those regions in IGV to have an intuitive result (Fig. 3).

The entire phasing process (Supplementary Fig. 2) starts from the input of raw reads from ONT. The state-of-the-art basecallers, Bonito[36], Guppy[33], or Dorado[37] from ONT, can basecall raw ONT reads with methylation information included. We then used minimap2 v2.24[62] to align reads to the reference and keep the MM and ML tags. Clair3 v0.1-r12[63] was used for variant calling with the consideration of different read types and fed the variants into HapCUT2[4] (extractHAIRS --bam BAM --vcf VCF --ref REF --ont 1 --out FRAGMENT; hapcut2 --outvcf 1 --f FRAGMENT --VCF VCF--o OUTPUT) or WhatsHap (whatshap phase -o VCF --reference=REF input.vcf input.bam) for SNV phasing. We also filtered the allele frequency (AF > 0.25) and limited our interested region to "high confidence" regions provided by GIAB to remove potential Clair3's false positive calls. The WhatsHap *stats* module was used to visualize and generate readable GTF files containing SNV phase blocks. The WhatsHap *haplotag* module was applied to tag reads based on HapCUT2's SNV phasing results. MethPhaser takes WhatsHap haplotagged reads, HapCUT2 phased SNVs as input and performs methylation-based read phasing and SNV-phase block chaining. To check the accuracy of our phase block connection, we need to determine the true relationship between two neighboring SNV phase blocks. Two neighboring SNV phase blocks are considered to have the same haplotype assignment if their haplotype assignments' relationship against the truth VCF is the same. For example, if haplotype 1 in SNV phase block 1 is haplotype 2 in the truth VCF file while the SNV phase

block 2 has the same relationship, i.e., the haplotype 1 in SNV phase block 2 is also haplotype 2 in the truth VCF, we consider the SNV phase block 1 and SNV phase block 2 are having the same haplotype assignments. Vice versa, two neighboring SNV phase blocks' have switched relationships means their haplotype assignment relationships to the truth VCF are also switched. Given these truth relationship assignments, we can easily know the correctness of MethPhaser's each SNV phase block connection with our relationship output and further get the correct N50 on the HG002 sample in Fig. 2. WhatsHap *compare* was used on checking the flip error and switch error on HG002. The detailed parameters are listed in the Supplementary Table 1. We further assessed the switch error introduced by MethPhaser with the consideration of the phase block connections by dividing the number of new switches MethPhaser introduced with the number of the connections of the phase blocks established by MethPhaser. We report this number in Supplementary Data2.

### Statistics and reproducibility
All analyses were performed with the same parameters as the benchmarking section stated.

### Reporting summary
Further information on research design is available in the Nature Portfolio Reporting Summary linked to this article.

## Data availability
Source data of Fig. 2 for all available samples are stored in Supplementary Data 1. Source data for generating Fig. 3 is stored at Supplementary Data 2. Source data for generating Fig. 5 is stored in Supplementary Data 3 and 4. Source data for per-chromosome N50, switch error stats of HG002 with different read types and coverages is stored in Supplementary Data 8–10. The HG002 kit 10 R9 data used in this study is publicly available at https://labs.epi2me.io/gm24385-5mc-remora/, called by Bonito base caller with profile dna_r9.4.1_e8_sup@v3.3. The reference genome is hg38 from Genome in a Bottle (GIAB NIST). The reads are at the coverage of 80x, and to test the effectiveness of our method in lower coverages, we also randomly subsampled the reads into 60x and 30x. The HG002 kit 14 R10 data used in this study is publicly available at https://humanpangenome.org/data.html. It is sequenced with Oxford Nanopore kit 14 (400 bps speed) and pore version r10.4.1 and basecalled with Dorado v4.0.0 SUP model + Reomra. The reference genome is also hg38. The reads are at the coverage of 60X, and we also subsampled it into 30X. The HPRC sample data used in this dataset are publicly available at https://github.com/human-pangenomics/hpgp-data. The raw reads are re-basecalled with Dorado + Remora the R9.4.1 data (SUP model, 5mCG modifications). The reference genome is also hg38. The samples' coverages are various, around 60X. GIAB "high-confidence" region, also called Tier 1 region is provided by the GIAB website, which is available at https://www.nist.gov/programs-projects/genome-bottle. The HGSC001-004 block connection data generated in this study have been deposited in the Zenodo under the accession code https://doi.org/10.5281/zenodo.11195008. The raw HGSC001-004 data are protected and are not available due to data privacy laws.

## Code availability
MethPhaser was written in Python3. The software MethPhaser V0.0.1 that was used in this paper and the script that was used to generate results are available online through GitHub at https://github.com/treangenlab/methphaser under the MIT License.

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

## Acknowledgements

F.J.S. was part supported by NIH: UM1HG008898, 1U01HG011758-01, T.J.T. and Y.F. were supported in part by NIH NIAID P01AI152999. Y.F. was supported in part by the Ken Kennedy Institute Computer Science Engineering Enhancement Fellowship, funded by the Rice Oil & Gas HPC Conference.

## Author contributions

Y.F., S.A., and J.B. developed the initial concept; Y.F. developed the software performed the experiments, and analyzed the data from S.A. and F.S. M.M. performed the experiment on the blood sample. F.J.S., T.J.T. and Y.F. wrote the manuscript. S.J. supervised and funded the project. All authors discussed the results and approved the manuscript.

## Competing interests

Y.F. was an intern at Oxford Nanopore Technologies, Inc.. S.A., J.B., and S.J. are employees of Oxford Nanopore Technologies, Inc., and are stock or stock option holders of Oxford Nanopore Technologies plc. FJS receives research support from Oxford Nanopore, PacBio, Illumina, and

Genentech. The remaining authors declare no competing interests. Oxford Nanopore Technologies products are not intended for use for health assessment or to diagnose, treat, mitigate, cure, or prevent any disease or condition.
