## [Peer Review File · Nature Communications]

MethPhaser: methylation-based long-read haplotype phasing of human genomesEditorial Note: This manuscript has been previously reviewed at another journal that is not operating a transparent peer review scheme. This document only contains reviewer comments and rebuttal letters for versions considered at *Nature Communications*.

Reviewer #1 (Remarks to the Author):

I appreciate the authors' efforts in revision. The manuscript has been improved overall. I want to clarify that I do believe MethPhaser is technically correct and can improve phasing in some regions. The question is more on how much it improves on top of SNP-based phasing and whether INDEL/SV-based extension may be doing better. I am not sure the authors have addressed these questions well, but this point is less important.

1) The authors are using differential methylation to extend SNP-based phase blocks. This only works if long homozygous regions can be phased by differential methylation. In the first round of the review, I was asking the authors to show how often there are differential methylation signals when there are no heterozygous (het) variants. The authors instead only showed there are differential methylation signals -- these could just be caused by genomic variants. Note that I know methylation can differ between haplotypes. For example, if a SNP turns a CpG into another dinucleotide, it may change methylation at the same time. However, in this case, differential methylation does not have more power over het SNPs.

The authors added Fig 1a and 1b in revision. Fig 1a only shows patterns over a long distance. I can't tell how much this is only caused by het SNPs. In Fig 1b, there are three het SNPs in chr1:65,765,293-65,766,574. Two of them (chr1:65765625 and chr1:65766009) are mutating CpGs to non-CpGs. Contrary to the authors' intention, this example shows differential methylation is directly caused by SNPs. It is doing a disservice to the manuscript.

The authors also showed Fig 3 in the response. This figure is relevant to my request. However, the authors didn't show differential methylation in this region. In addition, in my original comment, I was requesting a genome-wide scan. I am not surprised to see differential methylation in some homozygous regions but the question is how often we see and whether they are dense and strong enough for reliable phasing. If the authors can see differential methylation in chr9:91,456,428-91,549,296, I would suggest they replace Fig 1b with this region. It would be good to show differential methylation for the HLA example in Fig 4 as well.

2) Reviewer 2 raised a critical point on how to report the error rates. I think the authors should report the switch error rates in MethPhaser-connected regions instead. I am showing how to calculate that below. There are ~159k het SNPs, or ~159k switches, in GIAB HG002. In Figure 3, on chr1, the switch error rate of SNP-only phasing is 0.00070526 and the error rate of MethPhaser phasing is 0.000713533. Then there are 112.1 vs 113.5 switch errors, respectively, for SNP-only and MethPhaser phasing. MethPhaser made one or two more errors. Let's say one error. From Supplementary Table "Comparison of wo GIAB high-conf", MethPhaser added 648 connections genome-wide. If these are evenly distributed along the genome, there are about ~54 connections on chr1. Then the switch error rate of MethPhaser is $1/54=2\%$. This is pretty good. Note that my calculation is approximate because the authors didn't give me exact counts.

3) Relatedly, the authors said "As we show for HG002, this is a minor 0.02% increase in error rate from 0.03% for SNV-based phasing to 0.05% improved phasing while gaining significantly longer phaseblocks." Where did 0.03% and 0.05% come from? I couldn't find them in the supplementary tables. Also, Figure 2 only keeps one significant digit for switch error rates. That is not enough: with 0.06% vs 0.07%, the MethPhaser error rate would be much higher than with 0.064% vs 0.066%. Please provide at least 3 significant digits and the exact number of switch errors.

4) The wording could be more accurate. For example, between line 395-397, the authors said "WhatsHap performs slightly better". A 46% increase in phased N50 is not "slightly"; MethPhaser only improves whatshap N50 by ~50%. The authors also said "the improvement of MethPhaser is similar compared to it on HapCUT2". In Supplementary Table 13, the improvement dropped from 2.0 folds to 1.5 folds with a much higher hamming error rate (1.60% vs 7.91%).

Reviewer #2 (Remarks to the Author):

In this revised version, the authors have incorporated additional statistical analyses to enhance the credibility of methylation-based phase block extension. However, the study still lacks crucial analyses that would convincingly demonstrate the efficacy and reliability of their approach.

1. The paper does not present a statistical analysis on the density and quantity of CpGs that exhibit significant allelic differential methylation states and could be utilized for phase block extension across the human genome.

2. The flipping (hamming) error rate and switch error rates for SNV phasing on ONT reads, as presented, raise concerns due to their low values (less than 0.1% in most instances). Such rates seem too low to be accepted without skepticism. While the flip and switch error rates reported via Whatsp appear largely comparable, crucial data on hamming distances remain absent in this revised version.

3. As methylation-based phasing principally serves to connect SNV-phased blocks, the error rate evaluations primarily reflect the SNV phasing rather than methylation-based phasing. It would be more appropriate to assess the quality of methylation-based phasing specifically in regions phased solely by methylation and compared them with SNV-based phasing.

Other Major Concerns: In Figure 4, there's an indication that SNV haplo-tagged reads were re-haplo-tagged post the application of MethPhaser. For example, within the central portion, the number of HP2 reads decreased following MethPhaser tagging. Regarding the distinctly marked somatic mutation, a pair of SNV haplo-tagged HP2 reads vanished after MethPhaser's application. The authors did not indicate that MethPhaser has the capacity to rectify SNV-based haplo-tagging, and based on theoretical understanding, it shouldn't. Thus, the veracity of Figure 4 is questionable.

Minor Revisions Needed: The phrasing in lines 217-218, specifically "Figure 2d illustrates the results, wherein MethPhaser can decrease the gap count from 81% (R10 30X coverage) to 54% (R9 60X coverage) of the prior SNV-based gap number", needs refinement.

Reviewer #3 (Remarks to the Author):

The revised manuscript is much improved and the authors' response to the comments from the previous review is good. I have the following additional comments:

1. The p-value threshold used for the wilcoxon rank test for identifying methylation sites is not given. Nor was there any discussion on how this threshold impacts the phasing. On the same point, no statistics were provided for the number of such sites identified across different genomes.

2. MethPhaser is able to phase large blocks of SNVs (with gaps between them) using methylation, therefore, the main error mode of MethPhaser should be "switch error" (i.e. long switches in phase) rather than flip errors (a single SNV is phased incorrectly relative to other SNVs). However, in Figure 2(c), the "MethPhaser Phasing Switch Error" is identical to the "SNV Phasing Switch Error" for many datasets while the "flip rate" is increased for all datasets. For the HG002 genome results (lines 193-197), the authors write: "This includes an increase in the flip rate of 0.02% and no switch error increase compared to the SNV phasing alone. The main error mode from MethPhaser is to assign blocks of phasing wrongly, resulting in switch errors or flipping."

I am quite confused by this. If the authors can clarify, it would be helpful.

3. Also, the phasing error rates in Figure 2 are presented for the full dataset. This makes it difficult to assess how many block pairs did MethPhaser phase incorrectly. Methphaser essentially phases a subset of haplotype

blocks that are not phased using SNVs. If the overall switch error rate increases from 0.06% to 0.07% (as an example), the error rate of the additional phasing would actually be much higher. It is important to provide statistics on the number of phasing errors introduced by MethPhaser relative to the number of additional blocks phased.

4. Line 328: No details about the 'experimental evaluation' were provided.

5. I did not understand why the phaseblock N50 for the HG002 genome used the trio-based VCF:

"On top of that, we excluded the regions we did not connect correctly to get the correctly connecting N50 with HG002 samples based on the comparison of the SNV-based phasing method and the trio-phased VCF provided by GIAB."

6. The authors' response to my comment about alternative approaches for extending sequence based haplotypes is not satisfactory. The problem has been studied in many previous papers. See, e.g. Kuleshov et al. <https://www.nature.com/articles/nbt.2833>. In this paper, LD patterns are used to extend haplotypes inferred from sequence reads. This improves the mean length of haplotypes from 60 kb to 600 kb (10-fold). Many statistical phasing algorithms (e.g. SHAPEIT4) can phase individual genomes using large reference panels and combine information from sequence reads and population LD information.

We thank the reviewers and editor for their constructive comments and suggestions. We have replied to the comments below and made the suggested changes to the manuscript. All these changes are highlighted in blue.

Reviewer #1 (Remarks to the Author):

I appreciate the authors' efforts in revision. The manuscript has been improved overall. I want to clarify that I do believe MethPhaser is technically correct and can improve phasing in some regions. The question is more on how much it improves on top of SNP-based phasing and whether INDEL/SV-based extension may be doing better. I am not sure the authors have addressed these questions well, but this point is less important.

We thank the reviewer for this important clarification. We think that SNV phasing is an easier signal, but as we tried to showcase in regions where tools (e.g. Whatsapp, Hapcut2) give up there is still a signal from methylation that can be leveraged.

1) The authors are using differential methylation to extend SNP-based phase blocks. This only works if long homozygous regions can be phased by differential methylation. In the first round of the review, I was asking the authors to show how often there are differential methylation signals when there are no heterozygous (het) variants. The authors instead only showed there are differential methylation signals -- these could just be caused by genomic variants. Note that I know methylation can differ between haplotypes. For example, if a SNP turns a CpG into another dinucleotide, it may change methylation at the same time. However, in this case, differential methylation does not have more power over het SNPs.

The authors added Fig 1a and 1b in revision. Fig 1a only shows patterns over a long distance. I can't tell how much this is only caused by het SNPs. In Fig 1b, there are three het SNPs in chr1:65,765,293-65,766,574. Two of them (chr1:65765625 and chr1:65766009) are mutating CpGs to non-CpGs. Contrary to the authors' intention, this example shows differential methylation is directly caused by SNPs. It is doing a disservice to the manuscript.

To this point it's worth highlighting that the phase blocks are formed by SNV phasing methods (Whatsapp or Hapcut2). They stop at these regions that we can put together using Methphaser either because there is a stretch of homozygosity or there is some conflicting information. In any case Methphaser is able to fill the gap across multiple regions. We had added Figure 1ab in the spirit because we thought the reviewer didn't believe methylation can be utilized for phasing in general and if we show a region without SNV phasing how can we demonstrate the correctness? Thus we decided to show an exemplar region to highlight that the methylation pattern follows the expected haplotype. We never claimed that phaseblocks that are reported by Whatsapp or Hapcut2 are wrong or that methylation has a stronger

signal than SNV in certain regions of the genome. Only that such phaseblocks can be extended and put together based on methylation signals.

The authors also showed Fig 3 in the response. This figure is relevant to my request. However, the authors didn't show differential methylation in this region. In addition, in my original comment, I was requesting a genome-wide scan. I am not surprised to see differential methylation in some homozygous regions but the question is how often we see and whether they are dense and strong enough for reliable phasing. If the authors can see differential methylation in chr9:91,456,428-91,549,296, I would suggest they replace Fig 1b with this region. It would be good to show differential methylation for the HLA example in Fig 4 as well.

We are seeing haplotype-specific methylations but in a sparse distribution. We here list several IGV plots across the reviewer suggested regions that contain the haplotype-specific methylation identified by MethPhaser.

Figure 1: haplotype-specific methylation in homozygous region. We here report 4 examples of haplotype-specific methylation suggested by MethPhaser in a homozygous region chr9:91,456,428-91,549,296.

As the reviewer requested, we have also attached several examples of haplotype-specific methylation in the unphased regions in Figure 4.

As the reviewer points out Fig 3 shows the N50 improvements while little to no extra error is added with respect to the phasing. Thus, basically demonstrating that these regions can be extended by methylation. We have extended the text by adding that *HLA-E* and *HLA-C* regions were connected in this way.

2) Reviewer 2 raised a critical point on how to report the error rates. I think the authors should report the switch error rates in MethPhaser-connected regions instead. I am showing how to calculate that below. There are ~159k het SNPs, or ~159k switches, in GIAB HG002. In Figure 3, on chr1, the switch error rate of SNP-only phasing is 0.00070526 and the error rate of MethPhaser phasing is 0.000713533. Then there are 112.1 vs 113.5 switch errors, respectively, for SNP-only and MethPhaser phasing. MethPhaser made one or two more errors. Let's say one error. From Supplementary Table "Comparison of wo GIAB high-conf", MethPhaser added 648 connections genome-wide. If these are evenly distributed along the genome, there are about ~54 connections on chr1. Then the switch error rate of MethPhaser is $1/54=2\%$. This is pretty good. Note that my calculation is approximate because the authors didn't give me exact counts.

We thank the reviewer highlighting this. We indeed just followed the report from WhatsHap. We actually had listed our accuracy of block connection in the supplementary table 2. The accuracy of MethPhaser across HG002 varies from 96.27% to 89.26% across different runs. The way that we calculate this is stated in the last bit of the method section of the paper, which is slightly different from what the reviewer suggested.

"To check the accuracy of our phaseblock connection, we need to determine the true relationship between two neighboring SNV phaseblocks. Two neighboring SNV

phaseblocks are considered to have the same haplotype assignment if their haplotype assignments' relationship against the truth VCF is the same. For example, if haplotype 1 in SNV phaseblock 1 is haplotype 2 in the truth VCF file while the SNV phaseblock 2 has the same relationship, i.e., the haplotype 1 in SNV phaseblock 2 is also haplotype 2 in the truth VCF, we consider the SNV phaseblock 1 and SNV phaseblock 2 are having the same haplotype assignments. Vice versa, two neighboring SNV phaseblocks' have switched relationships means their haplotype assignment relationships to the truth VCF are also switched. "

To accommodate the reviewer's request, we followed the reviewer's method for accuracy calculation and added those into the supplementary table 2 also (row "switch error based error"). The error rate of the MethPhaser in R9 and R10 30X and 60X reads varies from 1.3% to 4.5%. We have attached the table of interest into our response.

Sample		R10 30X	R9 30X	R10 60X	R9 60X
Methylation connected Phase Block Accuracy	High ACC Param	89.26%	91.64%	94.49%	95.48%
	Switch Error based Error	4.55%	2.73%	2.31%	1.30%

3) Relatedly, the authors said "As we show for HG002, this is a minor 0.02% increase in error rate from 0.03% for SNV-based phasing to 0.05% improved phasing while gaining significantly longer phaseblocks." Where did 0.03% and 0.05% come from? I couldn't find them in the supplementary tables. Also, Figure 2 only keeps one significant digit for switch error rates. That is not enough: with 0.06% vs 0.07%, the MethPhaser error rate would be much higher than with 0.064% vs 0.066%. Please provide at least 3 significant digits and the exact number of switch errors.

We have added the 4 digits as requested and updated the figure 2 also.

4) The wording could be more accurate. For example, between line 395-397, the authors said "WhatsHap performs slightly better". A 46% increase in phased N50 is not "slightly"; MethPhaser only improves whatshap N50 by ~50%. The authors also said "the improvement of MethPhaser is similar compared to it on HapCUT2". In Supplementary Table 13, the improvement dropped from 2.0 folds to 1.5 folds with a much higher hamming error rate (1.60% vs 7.91%).

This is just a summary sentence in the discussion and pertains to the performance between HapCUT2 and WhatsHap. Obviously we didn't want to give a wrong impression. We have now changed this:

“ We have also compared the impact on MethPhaser's performance with HapCUT2 and WhatsHap's SNV phasing. We observed improvements using MethPhaser for both phasing methods. (Supplementary table 13). “

Reviewer #2 (Remarks to the Author):

In this revised version, the authors have incorporated additional statistical analyses to enhance the credibility of methylation-based phase block extension. However, the study still lacks crucial analyses that would convincingly demonstrate the efficacy and reliability of their approach.

1. The paper does not present a statistical analysis on the density and quantity of CpGs that exhibit significant allelic differential methylation states and could be utilized for phase block extension across the human genome.

In HG002, we have run MethPhaser's statistical test on each phaseblocks. We have found 412,944 CpGs, which is 0.0163% of the lengths of all phaseblocks. This means that on average, every 6,135 bps there will be a CpG that has statistical differences between two haplotypes.

2. The flipping (hamming) error rate and switch error rates for SNV phasing on ONT reads, as presented, raise concerns due to their low values (less than 0.1% in most instances). Such rates seem too low to be accepted without skepticism. While the flip and switch error rates reported via WhatsHap appear largely comparable, crucial data on hamming distances remain absent in this revised version.

We respectfully don't follow the reviewers logic here. We recently published another paper in Nature Methods (doi: [10.1101/2023.01.12.523790](https://doi.org/10.1101/2023.01.12.523790)) that shows similar phasing error rates, which are independent of this current work. In fact this previous work was carried out by R9 not R10 flow cells which further improves the accuracy.

In this paper we reported Hamming distance, which is simply the sum of all errors. Flip and switch errors just quantify the errors further and provide further insights. That is why we chose them to be reported. We have now also extended Supplementary Table 2 as requested by the reviewer 1.

3. As methylation-based phasing principally serves to connect SNV-phased blocks, the error rate evaluations primarily reflect the SNV phasing rather than methylation-based phasing. It would be more appropriate to assess the quality of methylation-based phasing specifically in regions phased solely by methylation and compared them with SNV-based phasing.

We show that the phase blocks can be extended with little to no additional errors. We show that based on two different SNV based phasing methods (WhatsHap and HapCUT2). Reviewer 1 further suggested a way to provide methylation phasing focused error rates, but obviously we wanted to set the additional errors that are small in respect to the SNV phasing error rates. We have now also extended Supplementary Table 2 as requested by the reviewer 1.

Other Major Concerns: In Figure 4, there's an indication that SNV haplo-tagged reads were re-haplo-tagged post the application of MethPhaser. For example, within the central portion, the number of HP2 reads decreased following MethPhaser tagging. Regarding the distinctly marked somatic mutation, a pair of SNV haplo-tagged HP2 reads vanished after MethPhaser's application. The authors did not indicate that MethPhaser has the capacity to rectify SNV-based haplo-tagging, and based on theoretical understanding, it shouldn't. Thus, the veracity of Figure 4 is questionable.

We need to tag the reads based on the phasing information one time for SNV and one time based on methylation. We assume the reviewer is pointing out the peak in the middle of the HP2 figure. The review is correct about MethPhaser does not change the SNV based per-read haplotype assignment, but it can flip the entire phaseblock's reads' haplotype assignment when we infer the neighboring phaseblocks has switched haplotype assignment. In Figure 4, you can clearly see the read alignment pattern in HP2 is flipped up to HP1 since MethPhaser infers those two phaseblocks are in switched haplotype assignment. In fact, this region is the test data that we submitted along with the paper, and the reviewer can check the BAM file for further confirmation. We have made this now clearer in the caption.

Minor Revisions Needed: The phrasing in lines 217-218, specifically "Figure 2d illustrates the results, wherein MethPhaser can decrease the gap count from 81% (R10 30X coverage) to 54% (R9 60X coverage) of the prior SNV-based gap number", needs refinement.

We changed this sentence to: "Overall MethPhaser can reduce the gap number from 81% (R10 30x coverage) to 54% (R9, 60x coverage) compared to the previous SNV-based gaps between phaseblocks (see Figure 2d) "

Reviewer #3 (Remarks to the Author):

The revised manuscript is much improved and the authors' response to the comments from the previous review is good. I have the following additional comments:

We thank the reviewer for this recognition.

1. The p-value threshold used for the wilcoxon rank test for identifying methylation sites is not given. Nor was there any discussion on how this threshold impacts the phasing. On the same point, no statistics were provided for the number of such sites identified across different genomes.

We assume the reviewer means the number of CpG sites that show significant signal for phasing? We have included this now also based on another reviewer suggestion. Same as our response to reviewer #2 question 1.

2. MethPhaser is able to phase large blocks of SNVs (with gaps between them) using methylation, therefore, the main error mode of MethPhaser should be "switch error" (i.e. long switches in phase) rather than flip errors (a single SNV is phased incorrectly relative to other SNVs). However, in Figure 2(c), the "MethPhaser Phasing Switch Error" is identical to the "SNV Phasing Switch Error" for many datasets while the "flip rate" is increased for all datasets. For the HG002 genome results (lines 193-197), the authors write: "This includes an increase in the flip rate of 0.02% and no switch error increase compared to the SNV phasing alone. The main error mode from MethPhaser is to assign blocks of phasing wrongly, resulting in switch errors or flipping."

I am quite confused by this. If the authors can clarify, it would be helpful.

The reviewer is right. Our assumption was also that it impacts switch and not flip errors more often. However, the WhatsHap benchmark highlighted the flip errors more often. We have pasted below an example from the WhatsHap compare documentation, which shows one flip reported instead of two switches. Maybe that is the error that WhatsHap more often assigns in these regions that can be extended. (Figure made from here:

<https://whatsnap.readthedocs.io/en/latest/guide.html#whatsnap-compare>)

Switch and flip example:

```
A  B  C
0|1 0|1 0|1
0|1 0|1 0|1
0|1 1|0 1|0
1|0 0|1 1|0
1|0 0|1 1|0
```

The A to B comparison contains one switch, whereas A vs C contains one flip (two switches).

3. Also, the phasing error rates in Figure 2 are presented for the full dataset. This makes it difficult to assess how many block pairs did MethPhaser phase incorrectly. Methphaser essentially phases a subset of haplotype blocks that are not phased using SNVs. If the overall switch error rate increases from 0.06% to 0.07% (as an example), the error rate of the additional phasing would actually be much higher. It

is important to provide statistics on the number of phasing errors introduced by MethPhaser relative to the number of additional blocks phased.

Figure 2 shows the additional error that MethPhaser introduces which is minimal on a genome wide perspective while extending the N50 of the phaseblocks. Reviewer 1 suggested a way to have a more focused calculation which we now included: “the switch error based phasing error in supplementary table 2” We also extended Supplementary Table 2 with this information.

Sample		R10 30X	R9 30X	R10 60X	R9 60X
Methylation connected Phase Block Accuracy	Switch Error based Error	4.55%	2.73%	2.31%	1.30%

4. Line 328: No details about the 'experimental evaluation' were provided.

We are sorry about this mis formulation. We just meant a computational evaluation with respect to an assembly. This has now been clarified but was also explained in the followed sentence.

5. I did not understand why the phaseblock N50 for the HG002 genome used the trio-based VCF:

"On top of that, we excluded the regions we did not connect correctly to get the correctly connecting N50 with HG002 samples based on the comparison of the SNV-based phasing method and the trio-phased VCF provided by GIAB."

Thanks for pointing this out. We have now reformulated the sentence: “ The regions that are not correctly connected by MethPhaser should not be considered into N50 increase. Therefore on top of that, when we are calculating the N50 increase, we removed the larger phaseblocks that we incorrectly connected based on the comparison of the SNV-based phasing method and the trio-phased VCF provided by GIAB. “

6. The authors' response to my comment about alternative approaches for extending sequence based haplotypes is not satisfactory. The problem has been studied in many previous papers. See, e.g. Kuleshov et al. <https://www.nature.com/articles/nbt.2833>. In this paper, LD patterns are used to extend haplotypes inferred from sequence reads. This improves the mean length of haplotypes from 60 kb to 600 kb (10-fold). Many statistical phasing algorithms (e.g. SHAPEIT4) can phase individual genomes using large reference panels and combine information from sequence reads and population LD information.

The reviewer asked us to also mention HiC and other approaches which we did in the discussion. “This is in contrast to for example Hi-C or Strand-Seq²⁶ data sets that require additional work and sequencing. Nevertheless, if these extra data sets can be produced, they often lead to improved phasing results than long reads alone⁵⁴. It is worth noting that PacBio HiFi reads⁴³ also contain the same methylation signal, which means MethPhaser can also be potentially used on PacBio HiFi data sets. “

We appreciate of course the work that is done over population-based LD information. This can be a great tool for many common variants in the genome. Methphaser on the other hand doesn't rely on common or rare SNV information and is for free when sequencing samples with long reads where certain phasing of rare variants (e.g. HLA or other regions they occur) can be improved. We have now extended the sentences mentioned above with “Previous work also shows improvements in phasing of common variants utilizing population data.” and cited the suggested paper plus shapeit2 and PhaseMe.

Reviewer #1 (Remarks to the Author):

I like the new metrics and new figures the authors provided in the response. They are more informative than the results in the main text. The authors should incorporate the new numbers/figures into the main text.

1) Please replace Fig 2b with "Switch Error based" error rate in the spreadsheet. As is pointed out by all three reviewers, the overall switch error rate is dominated by SNP-based phasing and doesn't tell us how well MethPhaser performs. Although the 18.00% "Switched Error based" error rate on the "R9 80X" dataset is high (why did the authors only leave out this number in the response?), the accuracy on other datasets looks good.

2) Also about the switch error rate of "R9 80X", the previous spreadsheet shows "0.04%" for both "SNV Phasing Switch Error" and "MethPhaser Phasing Switch Error". In the new spreadsheet, this is changed to "0.0411%" and "0.0563%". 0.0563% can't be rounded to 0.04%. Which version is correct? What causes the change? Is the new number a typo? Note that the percentages for other datasets are consistent between versions. Only "R9 80X" isn't.

3) Similarly on line 198, I asked where 0.03% and 0.05% came from but the authors didn't explain. The two percentages are inconsistent with Fig 2b.

4) The authors still misunderstood my intention about Fig. 1b, so let me try again. MethPhaser connects SNP-based phase blocks in regions where there are no heterozygous SNPs. It is important to show differential methylation in such regions. The current Fig 1b is caused by SNPs. It doesn't help to explain the power of MethPhaser and misleads readers to believe differential methylation in such regions could look this clean. The real cases are messier, more like the four sites in the response (chr9:91,499,234, chr9:91,508,196, chr9:91,512,538 and chr9:91,546,813). The authors should show these examples as a main subfigure instead.

Reviewer #1 (Remarks on code availability):

The source code is available. The README seems good. I have not run the pipeline, though.

Reviewer #2 (Remarks to the Author):

The authors have addressed many of my initial concerns, significantly improving the manuscript. Their work on applying allelically differentially methylated CpGs to extend phased blocks is commendable and contributes valuably to the field. However, there are still several critical issues that need to be resolved before I can recommend acceptance.

1. Concerns about Methodological Accuracy: The claim of 'remaining 99.98% phasing accuracy' in the abstract is misleading and seems to overestimate the capabilities of MethPhaser. This figure does not accurately reflect the method's performance, especially when considering the switch/flipping error rates. These rates appear to be substantially underestimated. I concur with Reviewer 1's suggestion for an alternative method to assess phasing accuracy, which seems more appropriate for depicting MethPhaser's true performance and should be used to replace the switch/flipping error rates reported by the authors in many places in the manuscript.

2. In my previous comments, I raised doubts about the reported low switch error rates for SNV phasing. The authors' reference to their publication in 'Nature Methods' suggests that the inclusion of SVs significantly improved phasing quality, while only using SNVs generated 0.17-0.21% switch error rates. However, the current study solely used SNVs. It is crucial for the authors to provide a detailed description of their SNV-based phasing method to clarify this discrepancy.

3. The presentation in Figure 4 remains unclear and fails to address my previous concerns (Reviewer attached figures). Specifically:

- a. The figure lacks essential details on SNV alleles and methylation states, which are crucial for understanding the haplotype extension process.
- b. Read assigned to HP1 and HP2 in different phased blocks have no clear relationship and should be distinguished, otherwise the figure is very misleading.
- c. The illustration of reads in the green frame seems inconsistent with the described methodology. Clarification is needed on why only these three reads were flipped after applying MethPhaser. Were they haplotagged via SNVs in the second phased block? The read flipped here seem to be primarily overlapping with the third phased block.
- d. I suggest rearranging Panel b above Panel a for a clearer comparison of phased block regions before and after applying MethPhaser.

4. The statement in the second paragraph of the Discussion regarding the limitations of using methylation for phasing, particularly concerning human sex chromosomes, is inaccurate. The early developmental deactivation of chromosome X in females should not significantly impact methylation-assisted phasing. This point requires correction for factual accuracy.

In conclusion, while the manuscript has made significant strides, addressing these remaining issues is essential for its acceptance. I look forward to seeing the revised version, which I believe has the potential to make a substantial contribution to the field.

Haplotype Specific Insertion

<

Reviewer #3 (Remarks to the Author):

I have no further comments

Reviewer #3 (Remarks on code availability):

A readme file is provided with installation instructions. I did not install or test the code.

We thank the reviewers and the editor's feedback for the last revision. Overall, we feel that the comments were quite positive and the manuscript is improved over the previous rounds. We have now replied to the comments below and integrated the suggestions into the manuscript. Both replies and changes are marked in blue.

REVIEWER COMMENTS

Reviewer #1 (Remarks to the Author):

I like the new metrics and new figures the authors provided in the response. They are more informative than the results in the main text. The authors should incorporate the new numbers/figures into the main text.

We thank the reviewer for the feedback. We have now added our new data and figures to the main manuscript.

1) Please replace Fig 2b with "Switch Error based" error rate in the spreadsheet. As is pointed out by all three reviewers, the overall switch error rate is dominated by SNP-based phasing and doesn't tell us how well MethPhaser performs. Although the 18.00% "Switched Error based" error rate on the "R9 80X" dataset is high (why did the authors only leave out this number in the response?), the accuracy on other datasets looks good.

We thank the reviewer for noticing this. We have added a figure 2d with updated switch error based accuracy.

Updated Figure 2

The reason that the 80X error rate is higher is because of the much higher hamming distance SNP-based phasing method produced in 80X reads (more than 2% while others less than 1.5%). This very large hamming distance means there are a large portion of

SNVs being wrongly assigned in long SNV phaseblocks produced with 80X reads, which thus also impacts MethPhaser. By default MethPhaser gives the SNV phasing the benefit of the doubt. We now have added an explanation of this issue to the discussion.

We thank the reviewer for noticing the 80X data is missing in the response, we leave that out because it is not a standard flow cell output. It is included in the manuscript.

2) Also about the switch error rate of "R9 80X", the previous spreadsheet shows "0.04%" for both "SNV Phasing Switch Error" and "MethPhaser Phasing Switch Error". In the new spreadsheet, this is changed to "0.0411%" and "0.0563%". 0.0563% can't be rounded to 0.04%. Which version is correct? What causes the change? Is the new number a typo? Note that the percentages for other datasets are consistent between versions. Only "R9 80X" isn't.

We thank the reviewer for pointing this out this was a mistake on our part which accidentally counted the sex chromosomes into the evaluation. This has now been corrected to 0.0411% and 0.0563%.

3) Similarly on line 198, I asked where 0.03% and 0.05% came from but the authors didn't explain. The two percentages are inconsistent with Fig 2b.

We are sorry for the confusion, this was a statement we made with the old dataset. We now have updated figure 2b (now 2c with an updated figure from the last round of revision) with the new data with the new evaluation approach that reviewers have brought up, and remove this sentence.

4) The authors still misunderstood my intention about Fig. 1b, so let me try again. MethPhaser connects SNP-based phase blocks in regions where there are no heterozygous SNPs. It is important to show differential methylation in such regions. The current Fig 1b is caused by SNPs. It doesn't help to explain the power of MethPhaser and misleads readers to believe differential methylation in such regions could look this clean. The real cases are messier, more like the four sites in the response (chr9:91,499,234, chr9:91,508,196, chr9:91,512,538 and chr9:91,546,813). The authors should show these examples as a main subfigure instead.

We thank the reviewer for mentioning this, we have updated our figure 1b with a better example. It is in an unphased region in chr6:30,893,180-30,893,730 round HLA genes. Here, methylation signals are scattered in unphased reads and further clustered by MethPhaser. This 1b clearly shows that there are methylation signals in unphased regions that can be further phased by MethPhaser. We have made sure those are not directly

caused by SNV on the CpG sites.

Reviewer #1 (Remarks on code availability):

The source code is available. The README seems good. I have not run the pipeline, though.

Reviewer #2 (Remarks to the Author):

The authors have addressed many of my initial concerns, significantly improving the manuscript. Their work on applying allelically differentially methylated CpGs to extend phased blocks is commendable and contributes valuably to the field. However, there are still several critical issues that need to be resolved before I can recommend acceptance.

1. Concerns about Methodological Accuracy: The claim of 'remaining 99.98% phasing accuracy' in the abstract is misleading and seems to overestimate the capabilities of MethPhaser. This figure does not accurately reflect the method's performance, especially when considering the switch/flipping error rates. These rates appear to be substantially underestimated. I concur with Reviewer 1's suggestion for an alternative method to assess phasing accuracy, which seems more appropriate for depicting MethPhaser's true performance and should be used to replace the switch/flipping error rates reported by the authors in many places in the manuscript.

We are now reporting both the switch error corrected by the number of newly introduced blocks as it was suggested, in addition to the switch error itself to be comparable to the SNV phasing results. We have now also adopted the abstract:

“Across control samples, we extend the phase length N50 by almost 3-fold while only introducing 1.3% phasing error.”

2. In my previous comments, I raised doubts about the reported low switch error rates for SNV phasing. The authors' reference to their publication in 'Nature Methods' suggests that the inclusion of SVs significantly improved phasing quality, while only using SNVs generated 0.17-0.21% switch error rates. However, the current study solely used SNVs. It is crucial for the authors to provide a detailed description of their SNV-based phasing method to clarify this discrepancy.

We respectfully disagree. Our point was not referencing the inclusion of SV at all! We pasted below the initial reply. The nature methods paper highlights phasing of assemblies and demonstrates that long read based phasing works well. We have described in the methods since the first version that we are using WhatsHap and HapCUT2. These are established SNV based phasing methods that are published by other groups and state of the art methods utilized in hundreds of other publications. The assessment of the phasing error is also done by WhatsHap. WhatsHap reports the flip, switch error as well as the Hamming distance. All commands were listed in the Supplementary table 10 also for completeness.

Our previous reply: “We respectfully don't follow the reviewers logic here. We recently published another paper in Nature Methods (doi: [10.1101/2023.01.12.523790](https://doi.org/10.1101/2023.01.12.523790)) that shows similar phasing error rates, which are independent of this current work. In fact this previous work was carried out by R9 not R10 flow cells which further improves the accuracy.

In this paper we reported Hamming distance, which is simply the sum of all errors. Flip and switch errors just quantify the errors further and provide further insights. That is why we chose them to be reported. We have now also extended Supplementary Table 2 as requested by the reviewer 1. “

3. The presentation in Figure 4 remains unclear and fails to address my previous concerns (Reviewer attached figures). Specifically:

a. The figure lacks essential details on SNV alleles and methylation states, which are crucial for understanding the haplotype extension process.

We thank the reviewer's suggestion about adding details of methylation states and SNV alleles. Noisy methylation states in this large scale IGV image are impossible to read and will definitely lower the readability of the figure 4. However, we have added the illustration of SNV alleles.

b. Read assigned to HP1 and HP2 in different phased blocks have no clear relationship and should be distinguished, otherwise the figure is very misleading.

In this case, we want to argue that figure 4 is only a showcase of the MethPhaser's result in HLA regions, not an illustration of the methodology of MethPhaser.

c. The illustration of reads in the green frame seems inconsistent with the described methodology. Clarification is needed on why only these three reads were flipped after applying MethPhaser. Were they haplotagged via SNVs in the second phased block? The read flipped here seem to be primarily overlapping with the third phased block.

(Figure from reviewer)

Short answer yes the reads are tagged with the second phaseblock because they haven't reached the beginning of the third phaseblock yet. To increase the readability we have attached a zoomed-in SNV phaseblock at the bottom of figure 4 to clarify the boundary of the 3 SNV phaseblocks on the IGV plot.

Also, due to the noisy nature of methylation signals, MethPhaser might sometimes classify a read into the wrong haplotype when the methylation patterns on it are too noisy. However, this is very hard to check because the reads are unhaplotagged by SNV-based methods

d. I suggest rearranging Panel b above Panel a for a clearer comparison of phased block regions before and after applying MethPhaser.

We have now modified figure 4 incorporating this change.

To further showcase the strength of MethPhaser, we have also included a more detailed figure with methylation states around that haplotype-specific insertion (chr6:30,893,462-30,894,563) phased by MethPhaser. In this figure, we can see that MethPhaser identified and utilized haplotype-specific methylations to improve read phasing and variant phasing. This figure was added as supplementary figure S4.

4. The statement in the second paragraph of the Discussion regarding the limitations of using methylation for phasing, particularly concerning human sex chromosomes, is inaccurate. The early developmental deactivation of chromosome X in females should not significantly impact methylation-assisted phasing. This point requires correction for factual accuracy.

The X chromosome activation/ deactivation is a random process that occurs in the gastrulation stage (~3 weeks of the embryo). As this is a random process it does not follow the haplotype structure that distinguishes one copy of X from the other. Thus our methylation cannot be used in X chromosomes. We frankly wish this wasn't the case! The only time this complete random process is skewed and not completely random is in specific genetic disease cases. Here if there is a non-methylated gene that carries a

pathogenic mutation, it can lead to cell death and one can observe a skewing of this random process. Note this is due to cell death.

Belmont J. W. Genetic control of X inactivation and processes leading to X-inactivation skewing. *American Journal of Human Genetics* . 1996;58(6):1101–1108.

Migeon B. R., Haisley-Royster C. Familial skewed X inactivation and X-linked mutations: unbalanced X inactivation is a powerful means to ascertain X-linked genes that affect cell proliferation. *The American Journal of Human Genetics* . 1998;62(6):1555–1557. doi: 10.1086/301858

In conclusion, while the manuscript has made significant strides, addressing these remaining issues is essential for its acceptance. I look forward to seeing the revised version, which I believe has the potential to make a substantial contribution to the field.

We thank the reviewer for these remarks. We think this indeed improved the manuscript and its clarity.

Reviewer #3 (Remarks to the Author):

I have no further comments

Reviewer #3 (Remarks on code availability):

A readme file is provided with installation instructions. I did not install or test the code.

We got notified from the editor that some of our replies were not satisfactory, especially with regard to previous questions 1,2 and 6. Thus, we reformulated our replies and added additional information in the following responses:

Previous Q1. The p-value threshold used for the wilcoxon rank test for identifying methylation sites is not given. Nor was there any discussion on how this threshold impacts the phasing. On the same point, no statistics were provided for the number of such sites identified across different genomes.

We have now made this clearer in the manuscript method section: “We use the Wilcoxon rank sum test^{44,58} to determine whether the two allele’s base modification probabilities are statistically different (p value < 0.05).”. The p-value was a standard $p < 0.05$ threshold.

Previous Q2. MethPhaser is able to phase large blocks of SNVs (with gaps between them) using methylation, therefore, the main error mode of MethPhaser should be "switch error" (i.e. long switches in phase) rather than flip errors (a single SNV is phased

incorrectly relative to other SNVs). However, in Figure 2(c), the "MethPhaser Phasing Switch Error" is identical to the "SNV Phasing Switch Error" for many datasets while the "flip rate" is increased for all datasets. For the HG002 genome results (lines 193-197), the authors write: "This includes an increase in the flip rate of 0.02% and no switch error increase compared to the SNV phasing alone. The main error mode from MethPhaser is to assign blocks of phasing wrongly, resulting in switch errors or flipping." I am quite confused by this. If the authors can clarify, it would be helpful.

Our response in addition:

We think the previous confusion is brought up by the insufficient digit number after the decimal point and thus rounding, which makes the switch error seem constant. For example, our previous R9 30X, the switch error before and after MethPhaser was 0.12% and 0.12%, and the flip rate was 0.06% and 0.07%, respectively. However, these rates are actually 0.1160% vs 0.1177% (switch increase 0.0017%), and 0.0633% vs 0.0650% (flip increase 0.0017%). After making the data reach to the 4 digit after the decimal point we can see the increase of switch error and the flip rate is similar. We have now modified the sentence in the manuscript to explain the results more clearly. This table is part of the Supplementary Table 2

Sample	R10 30X	R9 30X	R10 60X	R9 60X	R9 80X	R10 60X WhatsHap	R10 60X No Filter
Switch Error Increase	0.0017%	0.0018%	0.0010%	0.0012%	0.0152%	0.0049%	-0.0015%
Flip Rate Increase	0.0017%	0.0018%	0.0009%	0.0012%	0.0150%	0.0048%	-0.0003%

Our previous response:

The reviewer is right . Our assumption was also that it impacts switch and not flip errors more often. However, the WhatsHap benchmark highlighted the flip errors more often. We have pasted below an example from the WhatsHapcompare documentation, which shows one flip reported instead of two switches. Maybe that is the error that WhatsHapmore often assigns in these regions that can be extended. (Figure made from here: <https://whatshap.readthedocs.io/en/latest/guide.html#whatshap-compare>)

Switch and flip example:

```

A   B   C
0|1 0|1 0|1
0|1 0|1 0|1
0|1 1|0 1|0
1|0 0|1 1|0
1|0 0|1 1|0

```

The A to B comparison contains one switch, whereas A vs C contains one flip (two switches).

Previous Q6. The authors' response to my comment about alternative approaches for extending sequence based haplotypes is not satisfactory. The problem has been studied

in many previous papers. See, e.g. Kuleshov et al. <https://www.nature.com/articles/nbt.2833>. In this paper, LD patterns are used to extend haplotypes inferred from sequence reads. This improves the mean length of haplotypes from 60 kb to 600 kb (10-fold). Many statistical phasing algorithms (e.g. SHAPEIT4) can phase individual genomes using large reference panels and combine information from sequence reads and population LD information.

Our response in addition:

Another point maybe worth highlighting is that MethPhaser can be also utilized in minorities where maybe not enough population data is available to robustly infer LD. In addition MethPhaser can be applied also to other organisms such as non-model organisms where LD phasing is not possible due to lack of population data. We have now added this point to the discussion section.

Our previous response:

The reviewer asked us to also mention HiC and other approaches which we did in the discussion. “This is in contrast to for example Hi-C or Strand-Seq²⁶ data sets that require additional work and sequencing. Nevertheless if these extra data sets can be produced they often lead to improved phasing results than long reads alone⁵⁴. It is worth noting that PacBio HiFi reads⁴³ also contain the same methylation signal, which means MethPhaser can also be potentially used on PacBio HiFi data sets. “

We appreciate of course the work that is done over population based LD information. This can be a great tool for many common variants in the genome. Methphaser on the other hand doesn't rely on common or rare SNV information and is for free when sequencing samples with long reads where certain phasing of rare variants (e.g. HLA or other regions they occur) can be improved.

We have now extended the sentences mentioned above with “Previous work also shows improvements in phasing of common variants utilizing population data.” and cited the suggested paper plus shapeit2 and PhaseMe.

Reviewer #1 (Remarks to the Author):

I thank the authors for their responses, which are overall satisfactory. I still have one minor comment. The abstract said "Across control samples, we extend the phase length N50 by almost 3-fold while introducing minimal phasing errors (1.3%)". First, they cherry-picked the best data point "R90 60X". The error rate of "R90 80X" is ~17.6% in the newly added Figure 2d, which is quite high. Second, even for "R90 60X", MethPhaser only improved N50 by 133% ($=3.997/1.706-1$), not "3-fold". I would say something like "For a sample with ultra-long data, we increase the phase length N50 by XX-133% at phasing accuracy of 83.4-98.7%".

Reviewer #2 (Remarks to the Author):

Though the authors have addressed most of my concerns, their claim on the inability of using methylation for helping phase human X chromosomes is inappropriate. From a Nature Education article [<https://www.nature.com/scitable/topicpage/genetic-imprinting-and-x-inactivation-1066/>]: 'As originally proposed by Elizabeth Lyon, the selection of which X chromosome is inactivated is random, but after inactivation takes place, all the descendants of that cell are inactivated in the same way with respect to their X chromosomes.' In the study by Gershman et al. (Science, 2022), it denoted that 'the clonal female lymphoblast cell line GM12878, in which the Xi is always the paternal allele.' Therefore, for cell lines that derived from single cells, MethPhaser could help phase the X chromosomes. It should also be mentioned that for organisms with skewed X inactivation, methylation theoretically can help phasing.

Reviewer #3 (Remarks to the Author):

The revised manuscript (and abstract) is much better than before. I have no further comments.

Reviewer #3 (Remarks on code availability):

I was able to download the code and run it on the test data provided.

Reviewer #1 (Remarks to the Author):

I thank the authors for their responses, which are overall satisfactory. I still have one minor comment. The abstract said "Across control samples, we extend the phase length N50 by almost 3-fold while introducing minimal phasing errors (1.3%)". First, they cherry-picked the best data point "R90 60X". The error rate of "R90 80X" is ~17.6% in the newly added Figure 2d, which is quite high. Second, even for "R90 60X", MethPhaser only improved N50 by 133% ($=3.997/1.706-1$), not "3-fold". I would say something like "For a sample with ultra-long data, we increase the phase length N50 by XX-133% at a phasing accuracy of 83.4-98.7%".

We thank the reviewer for this final comment. However, we'd like to kindly point out that the R9 60X read is not the ultra-long reads that we mentioned in the main text, the ones in Figure 5 are the ultra-long reads. Nevertheless, we think the reviewer's suggestion about the accuracy is correct. We now have changed the abstract into

" For ONT R9 and R10 cell line data, we increase the phase length N50 by 78%-151% at a phasing accuracy of 83.4-98.7%"

Reviewer #2 (Remarks to the Author):

Though the authors have addressed most of my concerns, their claim on the inability of using methylation for helping phase human X chromosomes is inappropriate. From a Nature Education article [<https://www.nature.com/scitable/topicpage/genetic-imprinting-and-x-inactivation-1066/>]: 'As originally proposed by Elizabeth Lyon, the selection of which X chromosome is inactivated is random, but after inactivation takes place, all the descendants of that cell are inactivated in the same way with respect to their X chromosomes.' In the study by Gershman et al. (Science, 2022), it denoted that 'the clonal female lymphoblast cell line GM12878, in which the Xi is always the paternal allele.' Therefore, for cell lines that derived from single cells, MethPhaser could help phase the X chromosomes. It should also be mentioned that for organisms with skewed X inactivation, methylation theoretically can help phasing.

We thank the review for the detailed comment. We agree on the reviewer's point on the cell line. However the ultimate goal of MethPhaser is to be applied on real clinical samples, which also is the reason why we included a blood sample test in our paper. We now included this claim about cell lines in the discussion section along with the claim about the skewed X inactivation.

Our addition to discussion:

“The utilization of methylation for phasing is not without limitations, the most obvious of which is the inability to improve phasing for human sex chromosomes in clinical samples. This is because the random deactivation of chromosome X in females would lead to an inconsistent haplotype pattern, where the human cell line is an exception since the X chromosome is always the paternal allele⁵⁷, also the organisms with skewed X inactivation can theoretically benefit by MethPhaser. ”

Reviewer #3 (Remarks to the Author):

The revised manuscript (and abstract) is much better than before. I have no further comments.

Reviewer #3 (Remarks on code availability):

I was able to download the code and run it on the test data provided.